# OpenReview forum: "A Dendritic-Inspired Network Science Generative Model for Topological Initialization of Connectivity in Sparse Artificial Neural Networks"
_ICLR.cc/2026/Conference — Submitted to ICLR 2026_

### Official Review · Reviewer_83oV · 2025-10-16

**Soundness:** 3
**Presentation:** 3
**Contribution:** 3
**Rating:** 6
**Confidence:** 3

**Summary:**

This paper introduces the Dendritic Network Model (DNM), a parametric framework for generating network topologies to initialize neural architectures that can be trained in either static or dynamic sparse training setups. Drawing inspiration from neuroscience, the framework is evaluated across multiple architectures (MLPs and Transformers) through an extensive set of experiments spanning image classification and machine translation tasks. The experimental results suggest that the proposed methodology outperforms competing approaches.

**Strengths:**

A significant merit of this work is its exploration of an underrepresented research direction in the AI community, attempting to transfer knowledge from neuroscience to artificial intelligence. The paper addresses interesting scientific questions and supports its claims through a comprehensive experimental evaluation.

**Weaknesses:**

The paper overlooks a related line of research that could enhance the related works section: artificial neural networks with complex topologies [1,2,3,4]. In a sense, these studies can be considered a generalization of what is proposed here, as they are not limited to architectures constrained to bi/multipartite topologies (i.e., layered MLPs).

**Major**

Extreme Sparsity Regime: The presented results primarily focus on an extreme sparsity regime (99%). I wonder whether such a low number of connections might push the tested topologies into degenerate cnfigurations where the topological characteristics of each graph family could be at least partially compromised or diminished.

Hyperparameter Optimization and Fair Comparison: A more significant concern relates to how the results in the tables are constructed. Each result is presented as mean ± standard deviation over 3 different seeds. However, according to the appendix, if I understand correctly, the parameters selected for the DNM configurations appear to be the result of hyperparameter optimization. I am concerned that this may constitute a form of cherry-picking if the same level of hyperparameter search is not applied to the baseline initializations.

In other words, suppose the results presented for a specific task represent the 3 best networks from a sweep of 30 configurations. Would DNM still demonstrate superiority if we also selected the 3 best random networks from a sweep of 30 configurations?

Dense Baseline Comparison: I am curious about the performance of a dense MLP with the same number of connections as the tested sparse networks but with fewer neurons across the various experiments. I believe this additional baseline could make the significance of the presented results more convincing, especially given that current ML frameworks (both software and hardware) are particularly optimized for operations on dense MLPs.

**Questions:**

I would greatly appreciate comments addressing the major weaknesses reported above. I believe resolving these concerns could eliminate potential misunderstandings and lead to a more objective and informed evaluation of this work.

[1] R. L. S. Monteiro et al. “A Model for Improving the Learning Curves of Artificial Neural Networks”. en. In: PLOS ONE 11.2 (2016)

[2] S. Xie et al. “Exploring Randomly Wired Neural Networks for Image Recognition”. In: Proceedings of the IEEE/CVF International Conference on Computer Vision (ICCV). (2019)

[3] J. Stier and M. Granitzer. “deepstruct – linking deep learning and graph theory”. In: Software Impacts 11 (2022)

[4] T. Boccato et al. “Beyond multilayer perceptrons: Investigating complex topologies in neural networks”. In: Neural Networks 171 (2024)

---

> ### Author Response · Authors · 2025-12-03
>
> **Weakness 1: The paper overlooks a related line of research that could enhance the related works section: artificial neural networks with complex topologies [1,2,3,4]. In a sense, these studies can be considered a generalization of what is proposed here, as they are not limited to architectures constrained to bi/multipartite topologies (i.e., layered MLPs). [Reply 1/2]**
>
> We thank the reviewer for pointing out this important and relevant line of research. We revised the manuscript to include a discussion of these works. This literature, provides the essential context of this paper, as it demonstrates that network topology itself is a critical factor and that models based on complex graph principles (like ER, BA, and WS) can outperform traditional architectures. The works by Xie et al. and Boccato et al. explore network generators based on classical random graph models. Our paper, however, introduces a novel network topology generation that is build on a different set of biologically-inspired principles (like dendritic branching and localized receptive fields) rather than traditional models. Therefore, our work is not a subset of this field, but a new contribution to it. We have updated our "Related Works" section to make this context clear, situating DNM as a novel generative model alongside the classical models explored in the cited papers.
>
> _Dynamic sparse training (DST) trains a neural network with a sparse topology that evolves throughout the learning process. The initial arrangement of the connections is a critical aspect of this framework. This starting structure determines the initial pathways for information flow and acts as the foundational scaffold upon which the network learns and evolves. A well-designed initial topology can significantly improve a model’s final performance and training efficiency, whereas a poor starting point can severely hinder its ability to learn effectively. The principal topological initialization approaches for dynamic sparse training are grounded in network science theory, where three basic generative models for monopartite sparse artificial complex networks are the Erdos-Renyi (ER) model (Erdos & Renyi, 1960), the Watts-Strogatz (WS) model (Watts & Strogatz, 1998), and the Barabasi-Albert (BA) model (Barabasi & Albert, 1999). Since the standard WS and BA models are not directly designed for bipartite networks, they were recently extended into their bipartite counterparts and termed as Bipartite Small-World (BSW) and Bipartite Scale-Free (BSF) (Zhang et al., 2024b), respectively. BSW generally outperforms BSF for dynamic sparse training (Zhang et al., 2024b). The Correlated Sparse Topological Initialization (CSTI) (Zhang et al., 2024a) is a featureinformed topological initialization method that considers the links with the strongest Pearson correlations between nodes and features in the input layer. SNIP (Lee et al., 2018) is a data-informed pruning method that identifies important connections based on their saliency scores, calculated using the gradients of the loss function with respect to the weights. Ramanujan graphs (Kalra et al.) are a class of sparse graphs that exhibit optimal spectral properties, making them suitable for initializing neural networks with desirable connectivity patterns. The Bipartite Receptive Field (BRF) network model (Zhang et al., 2025) generates networks with brain-like receptive field connectivity. This is the first attempt to mimic the structure of brain connections in a sparse network initialization model. Radix-Nets (Kepner & Robinett, 2019) offer a deterministic approach to ”de novo” sparsity, utilizing mixed-radix numeral systems and the Kronecker product to construct topologies that ensure path-connectedness and symmetry while facilitating asymptotic sparsity. Finally, dendritic Artificial Neural Networks (dANNs) (Chavlis & Poirazi, 2025) introduce a bio-inspired architecture that mimics the structured connectivity and restricted input sampling of biological dendrites (e.g., using Local Receptive Fields). Unlike traditional approaches that strive for class specificity, this architecture fosters mixed-selective neuronal responses._

---

> ### Author Response · Authors · 2025-12-03
>
> **Weakness 1: The paper overlooks a related line of research that could enhance the related works section: artificial neural networks with complex topologies [1,2,3,4]. In a sense, these studies can be considered a generalization of what is proposed here, as they are not limited to architectures constrained to bi/multipartite topologies (i.e., layered MLPs). [Reply 2/2]**
>
> _While the methods discussed above primarily focus on initializing layered or bipartite structures, a parallel line of research investigates Artificial Neural Networks (ANNs) with general complex topologies, unconstrained by multipartite restrictions. Monteiro et al. (2016) demonstrated that hybrid topologies combining scale-free and small-world properties, inspired by the C. elegans connectome, can significantly improve learning curves. Moving beyond manual architecture design, Xie et al. (2019) utilize random graph models (ER, BA, WS) to generate “randomly wired” networks that achieve competitive performance in image recognition. To facilitate the translation between arbitrary graph structures and neural models, Stier & Granitzer (2022) introduced the deepstruct framework. More recently, Boccato et al. (2024) provided a systematic comparison of these architectures, revealing that complex, non-layered topologies can outperform traditional Multilayer Perceptrons (MLPs) in high-difficulty tasks by potentially exploiting compositional sparsity._

---

> > ### Author Response · Authors · 2025-12-03
> >
> > **Weakness 2: Extreme Sparsity Regime: The presented results primarily focus on an extreme sparsity regime (99%). I wonder whether such a low number of connections might push the tested topologies into degenerate configurations where the topological characteristics of each graph family could be at least partially compromised or diminished.**
> >
> > We thank the reviewer for raising this critical point. The choice of the 99% sparsity regime was a deliberate methodological decision, not a limitation. There is already evidence (https://proceedings.iclr.cc/paper_files/paper/2024/file/c9ef471a579197c4ed99df2aa542ce97-Paper-Conference.pdf ) that ultra-sparse (around 1% connection retained) topologies can achieve a learning advantage against fully connected networks, a phenomenon that is termed the "ultra-sparse advantage". Furthermore, we point out the following reasons why 99% is a reasonable sparsity for our comparisons:
> >
> > - Sparsity Amplifies Topological Differences: The reviewer's intuition is that fewer connections might lead to 'degenerate' configurations. However, from a network science perspective, the opposite is true. As network density increases, all topologies (random, scale-free, modular) mathematically converge toward a complete graph. Their distinguishing structural features are 'washed out,' and the choice of initialization becomes less relevant. Conversely, at extreme sparsity, the specific placement of the few available connections is paramount. It is in this regime that the fundamental differences between a structured, clustered (DNM) network, a random (ER) network, and a small-world (BSW) network become most pronounced and have the greatest impact on information flow.
> >
> > - 99% Sparsity is an Informative Test: Our paper's central hypothesis is that a bio-inspired initial topology provides a structural advantage. To test this, we must evaluate it in a regime where the initial topology matters most. Our experiments showed that as networks become denser, the performance gap between all initialization methods narrows and converges. Therefore, 99% sparsity is not a degenerate case; it is the most stringent and informative regime to isolate and measure the true benefit of a topological prior.

---

> > > ### Author Response · Authors · 2025-12-03
> > >
> > > **Weakness 3: Hyperparameter Optimization and Fair Comparison: A more significant concern relates to how the results in the tables are constructed. Each result is presented as mean ± standard deviation over 3 different seeds. However, according to the appendix, if I understand correctly, the parameters selected for the DNM configurations appear to be the result of hyperparameter optimization. I am concerned that this may constitute a form of cherry-picking if the same level of hyperparameter search is not applied to the baseline initializations. [REPLY 1/2]**
> > >
> > > We thank the reviewer for raising this critical point regarding fair comparison and hyperparameter optimization. We acknowledge the concern and would like to clarify our methodology, which we believe demonstrates the robustness of our DNM model beyond task-specific 'cherry-picking. We first want to clarify that the baseline methods also underwent a hyperparameter search to ensure a fair comparison. As detailed in Appendix B.1, we searched the key hyperparameters for the baselines, such as the rewiring probability beta for BSW and the randomness parameter r for BRF. Our DNM search, while larger due to the model's flexible nature, was conducted in the same spirit. The primary defense against 'cherry-picking' is transferability, which we demonstrate in Appendix I . This experiment shows that the single best DNM configuration found for static sparse training on MNIST can be applied 'one-shot', with no re-tuning, to the more complex EMNIST, Fashion MNIST, and CIFAR-10 datasets . In these tests, this transferred topology still outperforms the other initialization methods, demonstrating that its structural advantage is generalizable and not the result of task-specific tuning. We applied this same 'one-shot' transfer principle in our Transformer experiments. As noted in Section 4.3 and the Table 7 caption, the DNM results for the large WMT17 dataset were obtained without any new parameter search; we directly transferred the best configuration found on the IWSLT14 task . The strong performance of this transferred model again supports its robustness. To further solidify this point, we will add the extra transferability tests. We expanded Appendix K to rigorously test the generalizability of our topological priors. We conducted a 'one-shot' transfer experiment, applying the optimal configuration found for MNIST, EMNIST, Fashion MNIST, and CIFAR-10 to the other datasets without re-tuning. Here is the newly expanded Appendix K:
> > >
> > > _A critical question for our generative model of network topology is whether its principles are generalizable across different tasks. To investigate this, we conducted a comprehensive transfer learning experiment to assess if an optimal topology discovered on one task could be effectively applied to others. This tests the hypothesis that the DNM can capture fundamental structural priors beneficial for a class of problems, thereby reducing the need for extensive hyperparameter searches on every new dataset._
> > >
> > > **_Experimental Design_**
> > >
> > > _We identified the best-performing DNM hyperparameter configuration from the static sparse training experiments for each of the four datasets: MNIST, Fashion MNIST, EMNIST, and CIFAR-10. We then performed a cross-transfer analysis where the optimal configuration for a source dataset was used to initialize MLP models for the other three target datasets. We compared these transferred topologies against baseline initialization methods and against the DNM models specifically tuned for the target task (”DNM (Fine-tuned)”)._
> > >
> > > **_Results_**
> > >
> > > _The results, summarized in Table 14, reveal two distinct trends in topological transferability. First, we observe high transferability among the simpler grayscale datasets. Regardless of whether the parameters are transferred from MNIST, EMNIST, or Fashion MNIST, the resulting DNM models consistently outperform the baseline initialization methods on the other grayscale targets. In these cases, the transferred performance is often comparable to the task-specific fine-tuned models. However, a significant performance gap emerges when transferring topologies from simpler tasks to the more complex CIFAR-10 dataset. As shown in Table 14, configurations optimized for MNIST, EMNIST, or Fashion MNIST yield poor performance when applied to CIFAR-10 (approximately 47-48% accuracy compared to 58.71% for the fine-tuned model). This suggests that the structural priors learned from simpler inputs are insufficient for the complexity of natural images. Consequently, for practical scenarios involving image classification with MLPs, we recommend utilizing the parametric configuration derived from CIFAR-10. This configuration acts as a more robust baseline for complex tasks. The specific parameters for this recommended configuration are detailed in Table 15._

---

> > > > ### Author Response · Authors · 2025-12-03
> > > >
> > > > **Weakness 3: Hyperparameter Optimization and Fair Comparison: A more significant concern relates to how the results in the tables are constructed. Each result is presented as mean ± standard deviation over 3 different seeds. However, according to the appendix, if I understand correctly, the parameters selected for the DNM configurations appear to be the result of hyperparameter optimization. I am concerned that this may constitute a form of cherry-picking if the same level of hyperparameter search is not applied to the baseline initializations. [REPLY 2/2]**
> > > >
> > > > _**Table 14:**_
> > > >
> > > > | Method | MNIST | Fashion MNIST | EMNIST | CIFAR10 |
> > > > | :--- | :---: | :---: | :---: | :---: |
> > > > | **FC** | 98.80±0.00 | 90.87±0.02 | 87.08±0.04 | 62.35±0.13 |
> > > > | **CSTI** | 98.11±0.03 | 88.55±0.18 | 84.74±0.06 | 52.60±0.25 |
> > > > | **SNIP** | 98.03±0.03 | 88.65±0.07 | 85.19±0.04 | 61.89±0.48 |
> > > > | **Random** | 95.58±0.03 | 86.76±0.05 | 78.42±0.26 | 54.75±0.15 |
> > > > | **BSW** | 97.27±0.05 | 87.87±0.10 | 82.92±0.05 | 56.26±0.04 |
> > > > | **BRF** | 97.28±0.03 | 87.78±0.14 | 82.88±0.02 | 54.86±0.08 |
> > > > | **Ramanujan** | 96.39±0.10 | 86.44±0.14 | 81.78±0.08 | 54.61±0.32 |
> > > > | **RadiX-Nets** | 97.06±0.12 | 88.02±0.05 | 82.65±0.11 | 50.90±0.23 |
> > > > | **dANN-R** | 96.10±0.11 | 86.52±0.01 | 80.64±0.11 | 51.57±0.23 |
> > > > | **DNM (Fine-tuned)** | 98.07±0.09 | 88.86±0.21 | 85.63±0.10 | 58.71±0.28 |
> > > > | **Transferred from MNIST** | 98.07±0.09 | 88.92±0.09 | 85.80±0.04 | 48.24±0.25 |
> > > > | **Transferred from Fashion MNIST** | 98.05±0.09 | 88.86±0.21 | 85.63±0.10 | 47.52±0.39 |
> > > >
> > > > _**Table 15:**_
> > > >
> > > > | Parameter | Value / Distribution |
> > > > | :--- | :---: |
> > > > | Sparsity ($s$) | 0.99 |
> > > > | Mean Dendrites | 3 |
> > > > | Dendritic Distribution | Fixed |
> > > > | Receptive window width | 1.0 |
> > > > | Receptive window width distribution | Spatial Inverse-Gaussian |
> > > > | Degree Distribution | Spatial Inverse-Gaussian |
> > > > | Synaptic Distribution | Spatial Inverse-Gaussian |
> > > > | Layer Border Wiring Pattern | Wrap-around |

---

> > > > > ### Author Response · Authors · 2025-12-03
> > > > >
> > > > > **Weakness 4: Dense Baseline Comparison: I am curious about the performance of a dense MLP with the same number of connections as the tested sparse networks but with fewer neurons across the various experiments. I believe this additional baseline could make the significance of the presented results more convincing, especially given that current ML frameworks (both software and hardware) are particularly optimized for operations on dense MLPs.**
> > > > >
> > > > > We thank the reviewer for this insightful suggestion. The question of how our large, sparse models compare to a smaller, dense MLP with an equivalent parameter count is indeed a valuable one for contextualizing our results. We agree that this baseline will make the significance of our topological findings more convincing, especially given the current optimization of ML frameworks for dense operations. We add the following section in our Appendix (Appendix P) that addresses this comparison:
> > > > >
> > > > > _To rigorously validate the structural advantage of the Dendritic Network Model (DNM), it is necessary to decouple the benefits of network topology from the benefits of parameter efficiency. While Table 1 compares DNM against the over-parameterized Fully Connected (FC) network, a more rigorous baseline is the Dense Equivalent (DE) architecture._
> > > > >
> > > > > _**Motivation:**_ _At 99% sparsity, a sparse network retains only 1% of the parameters of the original FC model. A critical question arises: Does the sparse topology provide an inductive bias that facilitates learning, or could a dense network with the same tiny parameter budget perform equally well? The
> > > > > Dense Equivalent model is constructed by reducing the hidden dimension size H of the MLP such that the total number of trainable parameters equals the non-zero parameter count of the sparse DNM model. If DNM outperforms the DE baseline, it confirms that the arrangement of connections (the dendritic topology) is superior to a fully connected structure of equal capacity. Conversely, if the DE model performs comparably, it would suggest that the performance is governed solely by the parameter count, negating the need for complex sparse initialization._
> > > > >
> > > > > _**Results **_: _Table 17 presents the comparison between the Dense Equivalent baseline and the sparse initializations at 99% sparsity for image classification on the CIFAR-10 dataset. To match the extreme parameter reduction of 99% sparsity, the layer widths of the dense network must be drastically reduced. This severely limits the width of the representation that can be propagated through the network. In contrast, other sparse initialization methods maintain the original wide layer dimensions but sparsify the connections. As shown in the results, DNM significantly outperforms the Dense Equivalent baseline across all datasets._
> > > > >
> > > > > **_Table 17:_**
> > > > >
> > > > > _Comparison of Static Sparse Training vs. Dense Equivalent (DE) architectures on the CIFAR-10 dataset. Models utilize the same number of trainable parameters (approx. 1% of the original FC network). The DE model uses reduced hidden dimensions to match the parameter count of the 99% sparse DNM. Results are averaged over 3 seeds ± standard error._
> > > > >
> > > > > | Model | **CIFAR10** |
> > > > > | :--- | :--- |
> > > > > | **FC** | 62.35±0.13 |
> > > > > | **CSTI** | 52.60±0.25 |
> > > > > | **SNIP** | 61.89±0.48 |
> > > > > | **DE** | 56.37±0.31 |
> > > > > | **Random** | 54.75±0.15 |
> > > > > | **BSW** | 56.26±0.04 |
> > > > > | **BRF** | 54.86±0.08 |
> > > > > | **Ramanujan** | 54.61±0.32 |
> > > > > | **RadiX-Nets** | 50.90±0.23 |
> > > > > | **dANN-R** | 51.57±0.23 |
> > > > > | **DNM** | **58.71±0.28** |

---

> > > > > > ### Author Response · Authors · 2025-12-03
> > > > > >
> > > > > > **MESSAGE TO AREA CHAIR**
> > > > > >
> > > > > > We sincerely thank all reviewers for their detailed evaluations and constructive feedback, which has helped us significantly improve the quality and clarity of our manuscript.
> > > > > >
> > > > > > We are grateful that the reviewers recognized the work as "timely and relevant" (R1) and "novel" (R1, R2, R3), highlighting that the "parameterization of sparse topologies is well motivated by dendritic networks" (R2). We are encouraged that the reviewers found our experiments to be "comprehensive" (R1) and "convincingly validate the proposed model" (R3) across multiple architectures. Reviewers also appreciated that the paper addresses "interesting scientific questions" (R4) by exploring the "link between biological structure and sparse AI architectures" (R1).
> > > > > >
> > > > > > To provide greater clarity on the revisions made to our paper and the extensive experiments conducted to address the reviewers' concerns, we have summarized the modifications below:
> > > > > >
> > > > > > **Additional Experiments & Baselines:**
> > > > > >
> > > > > > **Generalization to RL & Graphs:**   To address the request for non-visual/non-language benchmarks (R3, Question 5), we extended our evaluation to Reinforcement Learning (PatternBandit task) and Graph Neural Networks (synthetic node classification). Results (new Appendix Q) confirm that DNM generalizes effectively to these domains, outperforming both dense baselines and random sparse initializations at high sparsity levels (90-99%). This represents the first ever application of the SOTA dynamic sparse training method, CHTs, on RL tasks.
> > > > > >
> > > > > > **Training Dynamics & Convergence Analysis:** We conducted a deep dive into the training dynamics (Addressing R3, Weakness 3). By analyzing loss, accuracy, and average gradient norm curves, and utilizing the Area Across Epochs (AAE) metric, we demonstrated (new Appendix R) that DNM exhibits significantly healthier gradient flow and faster convergence compared to random initialization.
> > > > > >
> > > > > > **New Baselines (RadiX-Net & dANNs):** We implemented and compared DNM against RadiX-Nets (Robinett & Kepner, 2018) and the recent dANNs (Chavlis & Poirazi, 2025). Results confirm that DNM consistently outperforms these specialized topologies on image classification tasks (Addressing R2, Weakness 1).
> > > > > >
> > > > > > **Transferability Analysis:** We added Appendix K to demonstrate the generalizability of DNM. We show that topological priors learned on simple datasets (e.g., MNIST) can be transferred "one-shot" to other tasks without re-tuning, outperforming baselines (Addressing R2, Weakness 2; R4, Weakness 3).
> > > > > >
> > > > > > **Learning Rate Sensitivity:** We conducted a sensitivity analysis (new Appendix O) for MLPs, identifying a stable optimal learning rate (0.025–0.05) across datasets, validating our experimental methodology (Addressing R1, Weakness 2; R3, Weakness 3).
> > > > > >
> > > > > > **Dense Baseline Comparison**: We added a comparison against Dense MLPs with equivalent parameter counts to further contextualize the "ultra-sparse advantage" of our method (Addressing R4, Weakness 2).
> > > > > >
> > > > > > **LLM Validation:** To demonstrate impact and scalability, we provided validation perplexity results on OpenWebText using LLAMA-60M, showing DNM outperforms state-of-the-art initialization (BRF) at 70% sparsity (Addressing R2, Weakness 7).
> > > > > >
> > > > > > **Clarifications & Structural Revisions:**
> > > > > >
> > > > > > **Topology vs. Performance Analysis**: We added a new Appendix N that quantitatively analyzes the correlation between graph-theoretic metrics (modularity, path length, etc.) and model accuracy, moving beyond speculation to principled explanation (Addressing R1, Weakness 4).
> > > > > >
> > > > > > **Restructured Section 3**: We completely restructured Section 3 to clearly separate the Biological Inspiration (3.1), the Generative Algorithm (3.2), Parameter Specification (3.3), and Geometric Characterization (3.4) for better flow and clarity (Addressing R1, Weakness 2).
> > > > > >
> > > > > > **Biological Context**: We expanded the Introduction to better contextualize DNM within recent progress in connectomics (e.g., MICrONS), clarifying that DNM is a generative topological abstraction rather than a direct biological simulation (Addressing R1, Weakness 1).
> > > > > >
> > > > > > **Practical Guidelines:** We added a section on "Practical guidelines for DNM initialization", providing recommended configurations for MLPs and Transformers to facilitate adoption without extensive hyperparameter search (Addressing R3, Weakness 1).
> > > > > >
> > > > > > We believe these extensive revisions and additional benchmarks comprehensively address the concerns raised, firmly establishing the robustness and novelty of the Dendritic Network Model.

---

### Official Review · Reviewer_qrtc · 2025-10-30

**Soundness:** 3
**Presentation:** 2
**Contribution:** 3
**Rating:** 6
**Confidence:** 2

**Summary:**

The paper introduces the Dendritic Network Model (DNM), a biologically inspired generative framework for initializing sparse artificial neural networks. Inspired by dendritic processing in biological neurons, DNM defines sparse connectivity through parametric distributions of dendrites, receptive fields, and synapses. It enables flexible control over modularity, hierarchy, and degree heterogeneity, allowing the generation of diverse network topologies. Experiments on image classification (MNIST, Fashion-MNIST, EMNIST, CIFAR-10) and machine translation tasks (Multi30k, IWSLT14, WMT17) show that DNM consistently outperforms classical sparse initialization methods at extreme sparsity levels, improving both efficiency and accuracy.

**Strengths:**

1. The paper introduces a novel and biologically grounded generative framework for sparse network initialization.
2. Extensive experiments across multiple datasets and architectures convincingly validate the proposed model.
3. The analysis establishes a clear link between network topology and task complexity, offering theoretical insights into structure–function relationships.

**Weaknesses:**

I am not a researcher in this area, but I feel interested in this topic. I have several concerns:
1. How sensitive is DNM to hyperparameter choices, and how would one select optimal configurations for unseen tasks?
2. The model’s computational overhead and scalability on very large-scale networks are not thoroughly discussed.
3. The work focuses primarily on initialization; it lacks exploration of how DNM interacts with other training dynamics beyond sparsity.
4. Can DNM be effectively combined with pruning or regularization techniques for additional efficiency gains?
5. Does the proposed framework generalize to non-visual or non-language domains, such as reinforcement learning or graph-based tasks? Please discuss.

Hope that the authors could answer the above questions. Thanks!

**Questions:**

See weakness.

---

> ### Author Response · Authors · 2025-12-03
>
> **Weakness 1: How sensitive is DNM to hyperparameter choices, and how would one select optimal configurations for unseen tasks?**
>
> We thank the reviewer for raising this important point regarding the practicality and adoptability of the model. We agree that providing clear guidance for selecting hyperparameters on unseen tasks is essential to avoid extensive search costs. To address this, we have added a new section to the manuscript titled "Practical guidelines for DNM initialization." In this section, we synthesize our experimental findings to provide robust baseline configurations. Specifically, we have distilled optimal settings for MLPs (based on our CIFAR-10 results) and Transformers (based on our transfer learning results from IWSLT14 to WMT17). These configurations are designed to serve as strong starting points for future researchers working on similar image classification or machine translation tasks. Here is the newly created section:
>
> _**Practical guidelines for DNM initialization**_
>
> _To facilitate the adoption of the Dendritic Network Model (DNM) without the need for extensive hyperparameter search, we provide recommended baseline configurations for MLPs and Transformers. These settings are derived from our most robust experimental results: the CIFAR-10 configuration for image classification and the IWSLT14-to-WMT17 transferred configuration for machine translation._
>
> _**MLP for Image Classification**_
>
> _For Multi-Layer Perceptrons (MLPs) applied to image classification tasks, particularly those involving complex visual features, we recommend the configuration detailed in Table 15. This setup was found to be the most effective for the CIFAR-10 dataset, and proved to stably outperform other initialization baselines on diverse datasets._
>
> _**Transformers for Machine Translation**_
>
> _For Transformer architectures applied to machine translation, we recommend the configuration that successfully transferred from IWSLT14 to the large-scale WMT17 dataset 4.3. As shown in Table 16, this configuration uses a higher dendrite count (M = 7) compared to MLPs and employs spatial distributions to organize connectivity._
>
> _**Table 15:**_
>
> _Recommended DNM parameter configuration for practical image classification scenarios. This configuration corresponds to the optimal settings identified for CIFAR-10._
>
> | Parameter | Value / Distribution |
> |---|---|
> | Sparsity (s) | 0.99 |
> | Mean Dendrites | 3 |
> | Dendritic Distribution | Fixed |
> | Receptive window width | 1.0 |
> | Receptive window width distribution | Spatial Inverse-Gaussian |
> | Degree Distribution | Spatial Inverse-Gaussian |
> | Synaptic Distribution | Spatial Inverse-Gaussian |
> | Layer Border Wiring Pattern | Wrap-around |
>
> _**Table 16:**_
>
> _Recommended DNM configuration for Transformers on Machine Translation tasks (derived from IWSLT14 to WMT17 transfer)._
>
> | Parameter | Value / Setting |
> |---|---|
> | Sparsity ($s$) | 0.90 |
> | Mean Dendrites | 7 |
> | Dendritic Distribution | Fixed |
> | Receptive window width | 1.0 |
> | Receptive window width distribution | Spatial Gaussian |
> | Degree Distribution | Spatial Inverse-Gaussian |
> | Synaptic Distribution | Spatial Inverse-Gaussian |
> | Layer Border Wiring Pattern | Wrap-around |

---

> > ### Author Response · Authors · 2025-12-03
> >
> > **Weakness 2: The model’s computational overhead and scalability on very large-scale networks are not thoroughly discussed.**
> >
> > The time complexity is O(${N_{in}}$ x ${N_{out}}$). We report the runtime of the creation of the DNM masks for different sizes:
> >
> > | Sandwich Layer Size| Runtime |
> > |---|---|
> > |10,000x10,000| <1.8s |
> > |1,000x1,00| <0.1s |
> > | 500x500 | <0.05s |
> > |100x100  | <0.001s |
> >
> > **Weakness 3: The work focuses primarily on initialization; it lacks exploration of how DNM interacts with other training dynamics beyond sparsity.**
> >
> > We agree that the primary focus of our work is to introduce and validate DNM as a novel sparse topological initialization framework. Our central hypothesis is that a bio-inspired, structured initial topology provides a significant structural advantage over conventional uniformely random or other sparse initializations. To test this, our primary investigation focused on the most critical dynamic related to our contribution: the dynamics of sparsity itself.
> >
> > To address the lack of exploration of how DNM interacts with other training dynamics beyond sparsity, we have performed the following tests that compare our model with a random sparse initialization for static sparse training on MLPs for image classification on CIFAR-10. For these tests, we selected the best performing parametric configuration for image classification on CIFAR-10. DNM demonstrates superior training dynamics compared to random initialization, as proven from the picture below. This picture compares the accuracy, loss, and average gradient norm through training epochs of the DNM-initialized network compared to the random one: https://anonymous.4open.science/r/ICLR2026-3B7F/rndnm.png .
> >
> > - Gradient Norms: In the Random initialization, the gradient norm for deeper layers (e.g., Layer 2, represented by the blue line) remains suppressed and stagnant around 0.3–0.4. In contrast, the DNM initialization facilitates significantly healthier gradient flow, with Layer 2 norms rising steadily to about 0.6. This suggests that the dendritic topology mitigates the vanishing gradient problem in sparse networks better than random connections, allowing for more effective weight updates in downstream layers.
> >
> > - Loss and accuracy: The DNM model achieves a lower final training and validation loss compared to the Random model. Consequently, DNM reaches a higher final testing accuracy (≈ 60%) compared to the Random baseline (≈55%).
> >
> > - Convergence speed analysis. To quantify the convergence speed, we utilise the area across epochs AAE metric (https://openreview.net/pdf?id=iayEcORsGd). AAE integrates the area underneath the learning curve over the training epochs E. A higher AAE indicates that the model reached higher accuracy levels faster and sustained them throughout the training process. The DNM model achieves a 53.32 AAE measure, while the random model achieves 49.68. DNM yields a higher AAE by a margin of 3.64, confirming that it converges significantly faster.
> >
> > We report these findings in a newly created Appendix R.

---

> > > ### Author Response · Authors · 2025-12-03
> > >
> > > **Weakness 4: Can DNM be effectively combined with pruning or regularization techniques for additional efficiency gains?**
> > >
> > > We thank the reviewr for this insightful question. Yes, DNM can be effectively combined with pruning techniques. We detail this in the newly created Appendix S:
> > >
> > > **_Combination of DNM with Pruning and Regularization techniques_**
> > >
> > > _DNM performs well as an initialization method of DST frameworks, which inherently involves pruning links. DNM can also be combined effectively with regularization techniques, as proved by the tests on image classification, where models were trained with weight decay, which is one of the most popular forms of L2 regularization._
> > >
> > >
> > > **Weekness 5: Does the proposed framework generalize to non-visual or non-language domains, such as reinforcement learning or graph-based tasks? Please discuss.**
> > >
> > > We thank the reviewer for this insightful comment. We emphasize that the Dendritic Network Model (DNM) is proposed as a general, brain-inspired topological initialization for linear layers across various architectures. While our initial focus was on visual and language domains, we acknowledge the importance of testing generalization. To address this, we performed two additional experiments targeting Graph-Based Learning and Reinforcement Learning scenarios.These are the first examples of testing CHTs (https://arxiv.org/abs/2501.19107) on reinforcement learning, confirming its applicability to a broad range of tasks. We added a new appendix section (Appendix Q) to discuss these findings. In both tests, we have fixed a parametric configuration for DNM (all distributions fixed, 3 dendrites) without performing any hyperparameter tuning.
> > >
> > > - _**Reinforcement Learning: Pattern Recognition Bandit**_: _We evaluated DNM on a "PatternBandit" task, where an agent must detect a specific local signal pattern ([1, 1, 1, 1]) hidden within a noisy state vector to receive a reward (Range: -1.0 to 1.0). This task tests the network's ability to maintain local feature integration capabilities under sparsity. We compared a Dense baseline against Random and DNM initializations using CHTs at 90% and 95% sparsity. The results are shown below:_
> > >
> > > | Sparsity | Model Configuration | Mean Reward (averaged over 3 seeds) | vs Random Baseline |
> > > |---|---|---|---|---|---|
> > > | **95%** | Dense (Baseline) | 0.71 | - |
> > > | | Random + CHTs | 0.65 | - |
> > > | | **DNM + CHTs** | **0.72** | +10.77% |
> > > | **90%** | Dense (Baseline) | 0.71| - |
> > > | | Random + CHTs| 0.65 | - |
> > > | | **DNM + CHTs** | **0.75** | +15.39% |
> > >
> > > _DNM is capable of outperforming both the dense and the randomly initialized sparse networks, both at 90% and 95% sparsity._
> > >
> > > - _**Graph Neural Networks:**_ _Synthetic Node Classification. We applied DNM to the linear transformation layers of a Graph Convolutional Network (GCN) on a synthetic dataset with high informative feature density (85% informative features). We tested extreme sparsity regimes (99%, 95%, and 90%) to see if DNM prevents topological collapse. The results are shown below:_
> > >
> > > | Sparsity | Model Configuration | Mean Accuracy (Over 3 seeds) | vs Random Baseline |
> > > |---|---|---|---|
> > > | **99%** | Dense (Baseline) | 0.7544 | - |
> > > | | Random Sparse | 0.6433 | - |
> > > | | **DNM + CHTs** | **0.7294** | +13.38% |
> > > | **95%** | Dense (Baseline) | 0.7544 | - |
> > > | | Random Sparse | 0.7572 | - |
> > > | | **DNM + CHTs** | **0.7594** |+0.29% |
> > > | **90%** | Dense (Baseline) | 0.7544 | - |
> > > | | Random Sparse | 0.7561 | - |
> > > | | **DNM + CHTs** | **0.7628** | 0.89% |
> > >
> > > _Like before, we confirm that the principles of DNM generalize effectively to non-visual and non-language domains._

---

### Official Review · Reviewer_Mon2 · 2025-10-31

**Soundness:** 2
**Presentation:** 3
**Contribution:** 2
**Rating:** 0
**Confidence:** 2

**Summary:**

This paper, inspired by dendrites in the brain, proposes DNM, a parameterization of a broad class of sparse neural network layer topologies, using parameters sparsity, dendritic distribution, receptive field width distribution, degree distribution, synaptic distribution, and layer border wiring pattern. A given topology replaces a fully connected layer. The authors evaluate the performance of different DNM topologies when replacing hidden layers with DNM layers on various image classification tasks as well as in transformers for machine translation tasks. The authors find that DNM perform slightly better than alternative sparse methods across the board.

**Strengths:**

- The paper has a thorough set of experiments, comparing to four different sparse baselines, on four different images datasets, plus two machine translation tasks. They evaluate for static sparse training as well as multiple dynamic sparse training methods. They conduct a sensitivity analysis and analyze the structure of the best performing networks.
- The parameterization of sparse topologies is well motivated by dendritic networks in the brain, and is novel.

**Weaknesses:**

- The design of dendritic sparse topologies is not very novel — it seems to be similar to several prior parameterizations of sparsity, perhaps just with more tunable parameters (see Robinett and Kepner, 2018 as an example).
- The performance benefit of the DNM layers is very minimal
- Some of the evaluation is suspect. For example, the reported standard error of accuracy on MNIST are in the 0.01% range, which seems unusually narrow. When I've done experiments on MNIST in the past, the variance between runs is much higher than 0.01%.
- The experiments are lacking many other crucial details (see questions)
- The authors suggest that they choose the best DNM model for each experiment from a hyperparameter sweep. Thus, the higher performance could just be due to trying more random architectures for DNM than other architectures.
- There is no single DNM architecture that works the best. So for any new problem, a new hyperparameter sweep would be required, rather than a "one size fits all" sparse architecture that outperforms fully connected networks. As a result, I'm not sure there is much impact of having a high performing sparse architecture.
- These results seem to be of minor impact to the community. The authors do not discuss why these architectures should be impactful, or if there's any potential for them to be adopted into mainstream state of the art neural network architecture design.

**Questions:**

Q1. what is the architecture of the models for MNIST? how many parameters are in the models? How many parameters are in the sparse models vs the fully connected models?

Q2. what is the evaluation strategy for the DNM models and other baselines on MNIST? how many different DNM variants are tested? How is the best model selected — on a held-out validation set? It could be useful to see a table of training, validation, and test accuracies for the different models tested.

Q3. Why are the error levels so low for the MNIST and other experiments? Is there other work that finds similar error levels when training different initializations on MNIST?

---

> ### Author Response · Authors · 2025-12-03
>
> **Weakness 1: The design of dendritic sparse topologies is not very novel — it seems to be similar to several prior parameterizations of sparsity, perhaps just with more tunable parameters (see Robinett and Kepner, 2018 as an example) [REPLY 1/3]**
>
> We thank the reviewer for explicitly recognizing the novelty of our parameterization and the thoroughness of our experiments in the "Strengths" section.
>
> Regarding the concern raised in the "Weaknesses" section about the novelty of the design in relation to prior work, such as Robinett and Kepner (2018), we appreciate the opportunity to clarify the fundamental distinctions between our Dendritic Network Model (DNM) and other methods like the RadiX-Net family of topologies. While both approaches aim to initialize sparse networks "de novo", the generative principles differ significantly.
>
> The approach of Robinett and Kepner (2018) is rooted in algebraic graph theory. They generate deterministic topologies (RadiX-Nets) using mixed-radix numeral systems and Kronecker products of adjacency submatrices. Their primary design goal is to ensure graph-theoretic properties such as constant expansion, path-connectedness, and symmetry. In contrast, DNM is a generative framework inspired by biological morphology. Rather than relying on algebraic products, DNM constructs connectivity via parametric distributions (spatial/non-spatial) of specific biological components: dendritic branches, localized receptive fields, and synapses. Our focus is on modeling a dendritic-inspired sparse network topology that allows for the emergence of modular and hierarchical structures rather than enforcing deterministic symmetry.
>
> To substantiate these differences and strengthen our evaluation, we have revised the Introduction to the manuscript as follows:
>
> _Artificial neural networks (ANNs) have demonstrated remarkable potential in various fields; however, their size, often comprising billions of parameters, poses challenges for both economic viability and environmental sustainability. [...]
> Distinct from these bio-mimetic approaches, the work of (Robinett & Kepner, 2019) is rooted in algebraic graph theory. They generate deterministic topologies (RadiX-Nets) using mixed-radix numeral systems and Kronecker products of adjacency submatrices. Their primary design goal is to ensure graph-theoretic properties such as constant expansion, path-connectedness, and symmetry. In contrast, DNM is a generative framework inspired by biological morphology. Rather than relying on algebraic products, DNM constructs connectivity via parametric distributions (spatial/non-spatial) of specific biological components: dendritic branches, localized receptive fields, and synapses. Our focus is on modeling a dendritic-inspired sparse network topology that allows for the emergence of modular and hierarchical structures rather than enforcing deterministic symmetry._
>
> As we clarified, our work addresses a distinct gap in the literature compared to other bio-inspired approaches. We created a new figure (Figure 1) to clarify this aspect. The figure can be found here: https://anonymous.4open.science/r/ICLR2026-3B7F/new_dend.png
> The caption is the following:
>
> _Comparison of the Dendritic Network Model with traditional point-neurons and
> existing dendritic architectures. (a) From point-neurons to dendritic topology. Traditional artificial neuron models (top) function as point-integrators, summing all synaptic inputs globally without spatial differentiation. In contrast, the DNM (bottom) introduces a brain-inspired topology where synaptic inputs are organized into distinct dendritic branches. This structure allows the output neuron to process inputs as clustered groups. (b) Comparison with existing dendritic network models. The left panel illustrates “Dendritic-emulated network processing” as seen in works like Li et al. (2020) and Chavlis & Poirazi (2025). In these architectures, dendrites are often modeled as explicit computational nodes forming an intermediate layer between inputs and the soma (a multilayer approach). The right panel illustrates the proposed DNM (a dendritic-inspired bipartite network topological initialization). Unlike previous dendritic-inspired models that have a tree-like multilayer structure, DNM embeds dendritic properties directly into the bipartite network topology. It treats dendrites as distinct clusters of links within a bipartite graph, connecting the soma to consecutive batches of inputs. This allows the network to inherit dendritic structural advantages through topological initialization rather than architectural expansion._
>
> We compare the accuracy of our initialization technique with the one introduced by Chavlis and Poirazi for static and dynamic training on MLPs for image classification over 4 datasets. We use Chavlis & Poirazi’s model dANN-R for comparison, since it proved to be the best performing in our tests. Here is the comparison:

---

> > ### Author Response · Authors · 2025-12-03
> >
> > **Weakness 1: The design of dendritic sparse topologies is not very novel — it seems to be similar to several prior parameterizations of sparsity, perhaps just with more tunable parameters (see Robinett and Kepner, 2018 as an example) [REPLY 2/3]**
> >
> > **Static training of MLPs at 99% sparsity for image classification**
> >
> > | Model | mnist | fmnist | emnist | CIFAR10 |
> > | :--- | :--- | :--- | :--- | :--- |
> > | FC | 98.78 ± 0.02 | 90.88 ± 0.02 | 87.13 ± 0.04 | 62.85 ± 0.16 |
> > | :--- | :--- | :--- | :--- | :--- |
> > | dANN-R | 96.10 ± 0.11 | 86.52 ± 0.01 | 80.64 ± 0.11 | 51.57 ± 0.23 |
> > | DNM | 98.07 ± 0.09 | 88.86 ± 0.21 | 85.63 ± 0.10 | 58.71 ± 0.28 |
> >
> >
> >
> > **SET on MLPs at 99% sparsity for image classification**
> >
> > | Model | mnist | fmnist | emnist | CIFAR10 |
> > | :--- | :--- | :--- | :--- | :--- |
> > | FC | 98.78 ± 0.02 | 90.88 ± 0.02 | 87.13 ± 0.04 | 62.85 ± 0.16 |
> > | :--- | :--- | :--- | :--- | :--- |
> > | dANN-R | 97.95±0.10 | 88.91±0.02 | 85.47±0.09 | 57.44±0.09 |
> > | DNM | 98.67±0.04 | 89.66±0.05  | 87.32±0.11 | 64.47±0.17 |
> >
> > **RigL on MLPs at 99% sparsity for image classification**
> >
> > | Model | mnist | fmnist | emnist | CIFAR10 |
> > | :--- | :--- | :--- | :--- | :--- |
> > | FC | 98.78 ± 0.02 | 90.88 ± 0.02 | 87.13 ± 0.04 | 62.85 ± 0.16 |
> > | :--- | :--- | :--- | :--- | :--- |
> > | dANN-R | 98.54±0.05 | 89.44±0.05 | 86.81±0.04 | 62.03±0.06 |
> > | DNM | 98.74±0.06 | 90.22±0.02 | 87.35±0.15 | 65.58±0.13 |
> >
> > Furthermore, we created a new appendix section, namely Comparison of DNM with other dendritic networks, to compare these two models in depth. Here is the section’s content:
> >
> > _Previous work, like that of Chavlis & Poirazi (2025), has explored the integration of dendritic structures into artificial neural networks, demonstrating improvements in parameter efficiency and robustness. Our Dendritic Network Model (DNM) distinguishes itself through its comprehensive approach to modeling dendritic connectivity. While Chavlis & Poirazi (2025) primarily focuses on the functional aspects of dendrites within a tree-like multilayered structure (dendritic and somatic layers), DNM embeds dendritic properties directly into the bipartite network topology. DNM treats dendrites as distinct clusters of links within a bipartite graph, connecting the soma to consecutive batches of inputs. This allows the network to inherit dendritic structural advantages through topological initialization rather than architectural expansion 1._
> >
> > _Section 4.2 presents a direct comparison between DNM and the best performing dANN model. The results indicate that DNM consistently outperforms dANNs across various image classification tasks at extreme sparsity levels (99%). These tests were performed by substituting each sandwich layer in our original network with dANN’s three-layered subnetwork, ensuring that the total number of trainable parameters remained constant for a fair comparison. For completeness, we perform additional tests on the network originally proposed by Chavlis & Poirazi (2025). In particular, we take a network instantiated with three sequential dendritic blocks, following the structure: Input → (Dendrite → Soma) × 3 → Output. We set the width of the dendritic layers to twice the size of the input layer_ (${N_{dendrite}}$ _= 2 ×_ ${N_{in}}$). _Conversely, the somatic integration layers are fixed at a width_ ${N_{soma}}$) _= 256, and the synapses parameter is set to 128_. _We compare one of the models introduced by Chavlis & Poirazi (2025), dANN-R, which we found to be the best performing one, against a modified one in which we replace the connectivity to the dendritic layers with a DNM topology that maintains the same sparsity levels. The parameters chosen for the DNM topology were extracted from the best-performing configuration found in Section 4.2 for EMNSIT at 99% sparsity. The results, summarized in Table 13, indicate that the DNM-initialized model outperforms dANN-R, which is the model that we found to perform the best out of those proposed by Chavlis & Poirazi. This further underscores the effectiveness of DNM's topological approach in enhancing network performance._
> >
> > _Table 13: Comparison of the dANN models (Chavlis & Poirazi, 2025) with a modified version where DNM replaces the connectivity patterns in the dendritic layer (dANN-DNM)._
> >
> > | Model | MNIST | EMNIST | Fashion MNIST |
> > | :--- | :---: | :---: | :---: |
> > |dANN-R | 98.49±0.05 | 85.96±0.09 | 89.77±0.08 |
> > |dANN-LRF | 98.51±0.00 | 85.60±0.15 | 89.39±0.07 |
> > |dANN-GRF | 98.53±0.06 | 86.26±0.05 | 89.77±0.08 |
> > | **dANN-DNM** | 98.70 ± 0.05 | 86.64 ± 0.07 | 90.09 ± 0.04 |
> >
> > To further address your comment, we have implemented the RadiX-Net (Robinett and Kepner, 2018) topology and added it to our experimental benchmarks. The results, now included in the revised paper, demonstrate that DNM achieves superior performance on the tested datasets. Here are the updated tables of the results obtained:

---

> > > ### Author Response · Authors · 2025-12-03
> > >
> > > **Weakness 1: The design of dendritic sparse topologies is not very novel — it seems to be similar to several prior parameterizations of sparsity, perhaps just with more tunable parameters (see Robinett and Kepner, 2018 as an example) [REPLY 3/3]**
> > >
> > > **Static sparse training on MLPs for image classification at 99% sparsity***
> > >
> > > | Method | MNIST | Fashion MNIST | EMNIST | CIFAR10 |
> > > | :--- | :--- | :--- | :--- | :--- |
> > > | **FC** | 98.80±0.00 | 90.87±0.02 | 87.08±0.04 | 62.35±0.13 |
> > > | **CSTI** | 98.11±0.03 | 88.55±0.18 | 84.74±0.06 | 52.60±0.25 |
> > > | **SNIP** | 98.03±0.03 | 88.65±0.07 | 85.19±0.04 | 61.89±0.48 |
> > > | **Random** | 95.58±0.03 | 86.76±0.05 | 78.42±0.26 | 54.75±0.15 |
> > > | **BSW** | 97.27±0.05 | 87.87±0.10 | 82.92±0.05 | 56.26±0.04 |
> > > | **BRF** | 97.28±0.03 | 87.78±0.14 | 82.88±0.02 | 54.86±0.08 |
> > > | **Ramanujan** | 96.39±0.10 | 86.44±0.14 | 81.78±0.08 | 54.61±0.32 |
> > > | **RadiX-Nets** | 97.06±0.12 | 88.02±0.05 | 82.65±0.11 | 50.90±0.23 |
> > > | **dANN-R** | 96.10±0.11 | 86.52±0.01 | 80.64±0.11 | 51.57±0.23 |
> > > | **DNM** | **98.07±0.09** | **88.86±0.21** | **85.63±0.10** | **58.71±0.28** |
> > >
> > > **Dynamic sparse training with SET on MLPs for image classification at 99% sparsity***:
> > >
> > > | Method | MNIST | Fashion MNIST | EMNIST | CIFAR10 |
> > > | :--- | :--- | :--- | :--- | :--- |
> > > | **FC** | 98.80±0.00 | 90.87±0.02 | 87.08±0.04 | 62.35±0.13 |
> > > | **CSTI** | 98.40±0.06 | 89.96±0.07 | 86.70±0.10 | 65.31±0.16 |
> > > | **SNIP** | 98.66±0.02 | 90.43±0.08 | 87.13±0.02 | 63.45±0.14 |
> > > | **Random** | 98.16±0.06 | 89.17±0.15 | 86.03±0.12 | 62.80±0.24 |
> > > | **BSW** | 98.22±0.03 | 89.28±0.09 | 86.21±0.02 | 64.13±0.11 |
> > > | **BRF** | 98.56±0.03 | 89.58±0.11 | 86.21±0.11 | 64.40±0.25 |
> > > | **Ramanujan** | 98.08±0.03 | 88.72±0.11 | 85.89±0.04 | 62.28±0.15 |
> > > | **RadiX-Nets** | 98.37±0.08 | 89.33±0.08 | 86.15±0.09 | 55.91±0.13 |
> > > | **dANN-R** | 98.54±0.05 | 89.44±0.05 | 86.81±0.04 | 62.03±0.06 |
> > > | **DNM** | **98.67±0.04** | **89.66±0.05** | **87.32±0.11** | **64.47±0.17** |
> > >
> > > **Dynamic sparse training with RigL on MLPs for image classification at 99% sparsity***:
> > >
> > > | Method | MNIST | Fashion MNIST | EMNIST | CIFAR10 |
> > > | :--- | :--- | :--- | :--- | :--- |
> > > | **FC** | 98.80±0.00 | 90.87±0.02 | 87.08±0.04 | 62.35±0.13 |
> > > | **SNIP** | 98.76±0.05 | 90.50±0.06 | 87.30±0.04 | 63.31±0.25 |
> > > | **CSTI** | 98.77±0.02 | 90.19±0.03 | 87.28±0.06 | 60.59±0.46 |
> > > | **Random** | 98.66±0.27 | 89.88±0.04 | 87.18±0.07 | 64.13±0.11 |
> > > | **BSW** | **98.74±0.03** | 90.12±0.06 | 87.28±0.10 | 65.19±0.23 |
> > > | **BRF** | 98.18±0.03 | 89.79±0.02 | 87.05±0.14 | 63.55±0.78 |
> > > | **Ramanujan** | 98.37±0.04 | 89.78±0.12 | 86.82±0.09 | 64.57±0.10 |
> > > | **RadiX-Nets** | 98.44±0.05 | 90.10±0.18 | 86.85±0.06 | 64.57±0.10 |
> > > | **dANN-R** | 98.54±0.05 | 89.44±0.05 | 86.81±0.04 | 62.03±0.06 |
> > > | **DNM** | **98.74±0.06** | **90.22±0.02** | **87.35±0.15** | **65.58±0.13** |
> > >
> > > **Dynamic sparse training with CHTs on MLPs for image classification at 99% sparsity***:
> > >
> > > | Method | MNIST | Fashion MNIST | EMNIST | CIFAR10 |
> > > | :--- | :--- | :--- | :--- | :--- |
> > > | **FC** | 98.80±0.00 | 90.87±0.02 | 87.08±0.04 | 62.35±0.13 |
> > > | **CSTI** | 98.70±0.04 | 90.56±0.09 | 87.47±0.04 | 69.59±0.20 |
> > > | **Random** | 98.46±0.08 | 90.02±0.14 | 87.04±0.09 | 64.62±0.08 |
> > > | **BSW** | 98.45±0.03 | 90.22±0.07  | **87.14±0.03** | 67.16±0.03 |
> > > | **BRF** | 98.52±0.08 | 90.55±0.08 | 87.09±0.10 | 66.72±0.96 |
> > > | **Ramanujan** | 98.37±0.04 | 89.78±0.12 | 86.82±0.09 | 64.57±0.10 |
> > > | **RadiX-Nets** | 98.44±0.05 | 90.10±0.18 | 86.85±0.06 | 64.92±0.11 |
> > > | **DNM** | **98.59±0.03** | **90.57±0.10** | **87.14±0.09** | **68.52±0.03** |
> > >
> > > *For fairness, to perform this comparison, we substituted each of the sandwich layers in our network with Chavlis and Poirazi’s three-layered subnetwork of sizes x, 2x, and x respectively, where x is the size of the input. Then, to compensate for the size difference between the two models, we initialized the dANNs in a way such that the number of connections between networks is the same. However, we also modified the original model published by Chavlis and Poirazi for EMNIST classification introducing our DNM model at each dendritic layer to generate the topology of the connectivity.
> > >
> > > We attribute our model's superior performance to a structural limitation in the dANN framework: the inclusion of dendritic integration layers increases network depth, which in turn forces each sandwich layer to be significantly sparser.
> > >
> > > These additions prove that we are not incremental at the level of design, since our dendritic model is fundamentally different from all prior works. Furthermore, our model is neither incremental from the point of view of its performance, as it clearly outperforms Chavlis and Poirazi’s model for initializing a dendritic model.

---

> > > > ### Author Response · Authors · 2025-12-03
> > > >
> > > > **Weakness 2: The performance benefit of the DNM layers is very minimal**
> > > >
> > > > Thanks for this comment. To better contextualize the performance benefit of DNM, we have incorporated significant new comparisons and analyses. First, we have expanded our evaluation to include a direct comparison with recent dendritic-inspired architectures, specifically the work of Chavlis and Poirazi (2025) that has been published on Nature communication. By substituting the sandwich layers in our network with their subnetwork structure to ensure a fair, parameter-equivalent comparison, our results demonstrate that DNM consistently outperforms the dANN model.
> > > > Second, we expanded Appendix K to rigorously test the generalizability of our topological priors. We conducted a 'one-shot' transfer experiment, applying the optimal configuration found for MNIST, EMNIST, Fashion MNIST, and CIFAR-10 to the other datasets without re-tuning. Here is the newly expanded Appendix K:
> > > >
> > > > _A critical question for our generative model of network topology is whether its principles are generalizable across different tasks. To investigate this, we conducted a comprehensive transfer learning experiment to assess if an optimal topology discovered on one task could be effectively applied to others. This tests the hypothesis that the DNM can capture fundamental structural priors beneficial for a class of problems, thereby reducing the need for extensive hyperparameter searches on every new dataset._
> > > >
> > > > _**Experimental Design**_
> > > >
> > > > _We identified the best-performing DNM hyperparameter configuration from the static sparse training experiments for each of the four datasets: MNIST, Fashion MNIST, EMNIST, and CIFAR-10. We then performed a cross-transfer analysis where the optimal configuration for a source dataset was used to initialize MLP models for the other three target datasets. We compared these transferred topologies against baseline initialization methods and against the DNM models specifically tuned for the target task (”DNM (Fine-tunedesg)”)._
> > > >
> > > > _**Results**_
> > > >
> > > > _The results, summarized in Table 14, reveal two distinct trends in topological transferability. First, we observe high transferability among the simpler grayscale datasets. Regardless of whether the parameters are transferred from MNIST, EMNIST, or Fashion MNIST, the resulting DNM models consistently outperform the baseline initialization methods on the other grayscale targets. In these cases, the transferred performance is often comparable to the task-specific fine-tuned models. However, a significant performance gap emerges when transferring topologies from simpler tasks to the more complex CIFAR-10 dataset. As shown in Table 14, configurations optimized for MNIST, EMNIST, or Fashion MNIST yield poor performance when applied to CIFAR-10 (approximately 47-48% accuracy compared to 58.71% for the fine-tuned model). This suggests that the structural priors learned from simpler inputs are insufficient for the complexity of natural images. Consequently, for practical scenarios involving image classification with MLPs, we recommend utilizing the parametric configuration derived from CIFAR-10. This configuration acts as a more robust baseline for complex tasks. The specific parameters for this recommended configuration are detailed in Table 15._
> > > >
> > > > _**Table 14:**_
> > > >
> > > > | Method | MNIST | Fashion MNIST | EMNIST | CIFAR10 |
> > > > | :--- | :---: | :---: | :---: | :---: |
> > > > | **FC** | 98.80±0.00 | 90.87±0.02 | 87.08±0.04 | 62.35±0.13 |
> > > > | **CSTI** | 98.11±0.03 | 88.55±0.18 | 84.74±0.06 | 52.60±0.25 |
> > > > | **SNIP** | 98.03±0.03 | 88.65±0.07 | 85.19±0.04 | 61.89±0.48 |
> > > > | **Random** | 95.58±0.03 | 86.76±0.05 | 78.42±0.26 | 54.75±0.15 |
> > > > | **BSW** | 97.27±0.05 | 87.87±0.10 | 82.92±0.05 | 56.26±0.04 |
> > > > | **BRF** | 97.28±0.03 | 87.78±0.14 | 82.88±0.02 | 54.86±0.08 |
> > > > | **Ramanujan** | 96.39±0.10 | 86.44±0.14 | 81.78±0.08 | 54.61±0.32 |
> > > > | **RadiX-Nets** | 97.06±0.12 | 88.02±0.05 | 82.65±0.11 | 50.90±0.23 |
> > > > | **dANN-R** | 96.10±0.11 | 86.52±0.01 | 80.64±0.11 | 51.57±0.23 |
> > > > | **DNM (Fine-tuned)** | 98.07±0.09 | 88.86±0.21 | 85.63±0.10 | 58.71±0.28 |
> > > > | **Transferred from MNIST** | 98.07±0.09 | 88.92±0.09 | 85.80±0.04 | 48.24±0.25 |
> > > > | **Transferred from Fashion MNIST** | 98.05±0.09 | 88.86±0.21 | 85.63±0.10 | 47.52±0.39 |
> > > > | **Transferred from EMNIST** | 98.05±0.09 | 88.86±0.21 | 85.63±0.10 | 47.52±0.39 |
> > > > | **Transferred from CIFAR-10** | 97.48±0.38 | 83.48±0.02 | 88.62±0.11 | 58.71±0.28 |
> > > >
> > > > _**Table 15:**_
> > > >
> > > > | Parameter | Value / Distribution |
> > > > | :--- | :---: |
> > > > | Sparsity ($s$) | 0.99 |
> > > > | Mean Dendrites | 3 |
> > > > | Dendritic Distribution | Fixed |
> > > > | Receptive window width | 1.0 |
> > > > | Receptive window width distribution | Spatial Inverse-Gaussian |
> > > > | Degree Distribution | Spatial Inverse-Gaussian |
> > > > | Synaptic Distribution | Spatial Inverse-Gaussian |
> > > > | Layer Border Wiring Pattern | Wrap-around |

---

> > > > > ### Author Response · Authors · 2025-12-03
> > > > >
> > > > > **Weakness 3: Some of the evaluation is suspect. For example, the reported standard error of accuracy on MNIST are in the 0.01% range, which seems unusually narrow. When I've done experiments on MNIST in the past, the variance between runs is much higher than 0.01%.**
> > > > >
> > > > > We thank the reviewer for their scrutiny regarding the reported standard error. To demonstrate the validity of our findings, we have analyzed the training, validation, and test accuracy trajectories of our best-performing MNIST model. Their curves can be found here:  https://anonymous.4open.science/r/ICLR2026-3B7F/ttv_acc.png .
> > > > >
> > > > > Furthermore, to provide full transparency regarding the variance, we have compiled the specific validation and test accuracies for each seed for our top 5 performing configurations. As seen in the table below, while there is minor fluctuation between seeds, the model consistently achieves high precision, resulting in the low standard error reported.
> > > > >
> > > > > | Rank | Mean Val | Seed 1 (Val / Test) | Seed 2 (Val / Test) | Seed 3 (Val / Test) |
> > > > > | :--- | :--- | :--- | :--- | :--- |
> > > > > | 1 | 98.6000 | 98.3000 / 97.9900 | 99.0000 / 98.2500 | 98.5000 / 97.9800 |
> > > > > | 2 | 98.5333 | 98.3000 / 98.1900 | 98.6000 / 98.0500 | 98.7000 / 98.1000 |
> > > > > | 3 | 98.4667 | 98.4000 / 98.1500 | 98.6000 / 98.0100 | 98.4000 / 97.8900 |
> > > > > | 4 | 98.4667 | 98.0000 / 98.1000 | 98.6000 / 97.9300 | 98.8000 / 98.0000 |
> > > > > | 5 | 98.4000 | 98.4000 / 98.0300 | 98.2000 / 98.0900 | 98.6000 / 98.1400 |
> > > > >
> > > > > Finally, we note that such stability is not unprecedented in recent literature using similar sparse topologies. For example, Zhang et al. (2025) (https://arxiv.org/abs/2501.19107) reported results with comparable standard error ranges.

---

> > > > > > ### Author Response · Authors · 2025-12-03
> > > > > >
> > > > > > **Weakness 4: The authors suggest that they choose the best DNM model for each experiment from a hyperparameter sweep. Thus, the higher performance could just be due to trying more random architectures for DNM than other architectures.**
> > > > > >
> > > > > > We appreciate the reviewer’s concern regarding the fairness of the comparison. We agree that if the performance gains were solely the result of an extensive random search, the optimal architectures would likely be overfitted to specific datasets and fail to generalize to new tasks without re-tuning.
> > > > > >
> > > > > > However, our results indicate that DNM captures fundamental structural priors rather than finding a "lucky" random initialization. We substantiate this with two key pieces of evidence from the manuscript where no hyperparameter sweep was performed on the target dataset:
> > > > > >
> > > > > > 1. Zero-Tuning Generalization on WMT17 (Transformers): As detailed in Section 4.3, for the large data scale WMT17 machine translation task, we explicitly did not perform a new hyperparameter search. Instead, we directly transferred the best-performing DNM configuration identified from the smaller IWSLT14 dataset . Despite this restriction, the DNM initialization outperformed the BRF baseline, demonstrating that the topological principles learned on a smaller task generalize to larger, more complex benchmarks without additional tuning.
> > > > > >
> > > > > > 2. Transferability of Optimal Topologies (MLPs): In Appendix K, we conducted a specific experiment to address this exact concern. We took the optimal hyperparameter configuration found for MNIST and applied it directly to initialize models for the other datasets. Even without task-specific tuning, this "Transferred DNM" outperformed Random, BSW, BRF, and Ramanujan initializations on EMNIST and Fashion MNIST. This confirms that the structural principles identified by DNM act as a powerful generalizable prior.
> > > > > >
> > > > > >
> > > > > > To further prove this point, we expanded our analysis to conduct a comprehensive cross-transferability experiment. We observed that structural priors generalize remarkably well among datasets of similar complexity: topologies transferred between MNIST, Fashion MNIST, and EMNIST consistently outperform baselines like Random, BSW, and BRF. However, we found that transferring priors from these simpler grayscale datasets to the more complex CIFAR-10 was suboptimal, highlighting a gap in structural requirements for processing natural images versus simpler patterns. Consequently, to provide practical value for diverse applications, we have explicitly identified and reported a robust "complex-task" configuration (derived from CIFAR-10) to serve as a recommended initialization for more challenging scenarios. Here is the final table for our transferability experiments:
> > > > > >
> > > > > > | Method | MNIST | Fashion MNIST | EMNIST | CIFAR10 |
> > > > > > | :--- | :---: | :---: | :---: | :---: |
> > > > > > | **FC** | 98.80±0.00 | 90.87±0.02 | 87.08±0.04 | 62.35±0.13 |
> > > > > > | **CSTI** | 98.11±0.03 | 88.55±0.18 | 84.74±0.06 | 52.60±0.25 |
> > > > > > | **SNIP** | 98.03±0.03 | 88.65±0.07 | 85.19±0.04 | 61.89±0.48 |
> > > > > > | **Random** | 95.58±0.03 | 86.76±0.05 | 78.42±0.26 | 54.75±0.15 |
> > > > > > | **BSW** | 97.27±0.05 | 87.87±0.10 | 82.92±0.05 | 56.26±0.04 |
> > > > > > | **BRF** | 97.28±0.03 | 87.78±0.14 | 82.88±0.02 | 54.86±0.08 |
> > > > > > | **Ramanujan** | 96.39±0.10 | 86.44±0.14 | 81.78±0.08 | 54.61±0.32 |
> > > > > > | **RadiX-Nets** | 97.06±0.12 | 88.02±0.05 | 82.65±0.11 | 50.90±0.23 |
> > > > > > | **dANN-R** | 96.10±0.11 | 86.52±0.01 | 80.64±0.11 | 51.57±0.23 |
> > > > > > | **DNM (Fine-tuned)** | 98.07±0.09 | 88.86±0.21 | 85.63±0.10 | 58.71±0.28 |
> > > > > > | **Transferred from MNIST** | 98.07±0.09 | 88.92±0.09 | 85.80±0.04 | 48.24±0.25 |
> > > > > > | **Transferred from Fashion MNIST** | 98.05±0.09 | 88.86±0.21 | 85.63±0.10 | 47.52±0.39 |
> > > > > > | **Transferred from EMNIST** | 98.05±0.09 | 88.86±0.21 | 85.63±0.10 | 47.52±0.39 |
> > > > > > | **Transferred from CIFAR-10** | 97.48±0.38 | 83.48±0.02 | 88.62±0.11 | 58.71±0.28 |

---

> > > > > > > ### Author Response · Authors · 2025-12-03
> > > > > > >
> > > > > > > **Weakness 5: There is no single DNM architecture that works the best. So for any new problem, a new hyperparameter sweep would be required, rather than a "one size fits all" sparse architecture that outperforms fully connected networks. As a result, I'm not sure there is much impact of having a high performing sparse architecture. [REPLY 1/2]**
> > > > > > >
> > > > > > > We appreciate the reviewer’s comment regarding the lack of a "one size fits all" architecture. However, we respectfully disagree that this limits the impact of the proposed method. In fact, our results suggest that the ability of DNM to adapt its topology to the specific complexity of the task is a significant theoretical and practical advantage. We address this concern through two main arguments:
> > > > > > >
> > > > > > > 1. Topology is a Task-Dependent Hyperparameter: Just as there is no "one size fits all" set of weights or hyperparameters (e.g., learning rate, depth, width) for artificial neural networks, our research demonstrates that there is no single sparse connectivity pattern that is optimal for every problem. In sparse training, tuning hyperparameters advocates for the proper learning of the connectivity shape suited to the data structure. Our analysis in Section 5 explicitly demonstrates that optimal topologies vary by task complexity, providing insights that a fixed "one size fits all" architecture would miss. To prove that there is no single sparse connectivity pattern that is optimal for every problem.
> > > > > > >
> > > > > > > 2. Robust Transferability (No "New Sweep" Required): While we agree that a task-specific sweep yields the absolute best performance, the reviewer’s concern that a new sweep is required for every new problem is incorrect. Our experiments demonstrate that DNM topologies act as robust structural priors that generalize well within a domain, as shown in our reply to Weakness 4.
> > > > > > > These results confirm that while task-specific tuning offers peak optimization, DNM provides a "one size fits class of problems" solution that is superior to fully random or rigid sparse initializations.
> > > > > > >
> > > > > > > In addition, we understand that it is important to provide a clear guidance for selecting hyperparameters on unseen tasks, to avoid extensive search costs. To address this, we have added a new section to the manuscript titled "Practical guidelines for DNM initialization." In this section, we synthesize our experimental findings to provide robust baseline configurations. Specifically, we have distilled optimal settings for MLPs (based on our CIFAR-10 results) and Transformers (based on our transfer learning results from IWSLT14 to WMT17). These configurations are designed to serve as strong starting points for future researchers working on similar image classification or machine translation tasks. Here is the newly created section:
> > > > > > >
> > > > > > > _**Practical guidelines for DNM initialization**_
> > > > > > >
> > > > > > > _To facilitate the adoption of the Dendritic Network Model (DNM) without the need for extensive hyperparameter search, we provide recommended baseline configurations for MLPs and Transformers. These settings are derived from our most robust experimental results: the CIFAR-10 configuration for image classification and the IWSLT14-to-WMT17 transferred configuration for machine translation._
> > > > > > >
> > > > > > > _**MLP for Image Classification**_
> > > > > > >
> > > > > > > _For Multi-Layer Perceptrons (MLPs) applied to image classification tasks, particularly those involving complex visual features, we recommend the configuration detailed in Table 15. This setup was found to be the most effective for the CIFAR-10 dataset, and proved to stably outperform other initialization baselines on diverse datasets._
> > > > > > >
> > > > > > > _**Transformers for Machine Translation**_
> > > > > > >
> > > > > > > _For Transformer architectures applied to machine translation, we recommend the configuration that successfully transferred from IWSLT14 to the large-scale WMT17 dataset 4.3. As shown in Table 16, this configuration uses a higher dendrite count (M = 7) compared to MLPs and employs spatial distributions to organize connectivity._
> > > > > > >
> > > > > > > **_Table 15_**
> > > > > > >
> > > > > > > _Recommended DNM parameter configuration for practical image classification scenarios. This configuration corresponds to the optimal settings identified for CIFAR-10._
> > > > > > >
> > > > > > > | Parameter | Value / Distribution |
> > > > > > > |---|---|
> > > > > > > | Sparsity (s) | 0.99 |
> > > > > > > | Mean Dendrites | 3 |
> > > > > > > | Dendritic Distribution | Fixed |
> > > > > > > | Receptive window width | 1.0 |
> > > > > > > | Receptive window width distribution | Spatial Inverse-Gaussian |
> > > > > > > | Degree Distribution | Spatial Inverse-Gaussian |
> > > > > > > | Synaptic Distribution | Spatial Inverse-Gaussian |
> > > > > > > | Layer Border Wiring Pattern | Wrap-around |

---

> > > > > > > > ### Author Response · Authors · 2025-12-03
> > > > > > > >
> > > > > > > > **Weakness 5: There is no single DNM architecture that works the best. So for any new problem, a new hyperparameter sweep would be required, rather than a "one size fits all" sparse architecture that outperforms fully connected networks. As a result, I'm not sure there is much impact of having a high performing sparse architecture. [REPLY 2/2]**
> > > > > > > >
> > > > > > > > **_Table 16:_**
> > > > > > > >
> > > > > > > > _Recommended DNM configuration for Transformers on Machine Translation tasks (derived from IWSLT14 to WMT17 transfer)._
> > > > > > > >
> > > > > > > > | Parameter | Value / Setting |
> > > > > > > > |---|---|
> > > > > > > > | Sparsity ($s$) | 0.90 |
> > > > > > > > | Mean Dendrites | 7 |
> > > > > > > > | Dendritic Distribution | Fixed |
> > > > > > > > | Receptive window width | 1.0 |
> > > > > > > > | Receptive window width distribution | Spatial Gaussian |
> > > > > > > > | Degree Distribution | Spatial Inverse-Gaussian |
> > > > > > > > | Synaptic Distribution | Spatial Inverse-Gaussian |
> > > > > > > > | Layer Border Wiring Pattern | Wrap-around |

---

> > > > > > > > > ### Author Response · Authors · 2025-12-03
> > > > > > > > >
> > > > > > > > > **Weakness 6:These results seem to be of minor impact to the community. The authors do not discuss why these architectures should be impactful, or if there's any potential for them to be adopted into mainstream state-of-the-art neural network architecture design.**
> > > > > > > > >
> > > > > > > > > We thank the reviewer for this comment and the opportunity to clarify the broader impact and practical relevance of our approach. Our goal is not to propose yet another specialized architecture, but to introduce a general sparse initialization scheme (DNM) that can be plugged into existing state-of-the-art models and training pipelines with minimal changes. In this sense, DNM is intended as an architectural primitive for sparse fully connected layers, rather than a standalone niche model.
> > > > > > > > >
> > > > > > > > > To better demonstrate its potential in mainstream settings, we have conducted additional experiments on large language models. In particular, we evaluate the state-of-the-art dynamic sparse training (DST) method CHTs on OpenWebText using LLaMA-60M with 70% sparsity and compare two initializations: BRF (the current SOTA initializer) and our DNM. We adopt the architecture and training setup of Zhang et al. (2025; https://arxiv.org/abs/2501.19107). The validation perplexities are:
> > > > > > > > >
> > > > > > > > > | Method | LLAMA-60M |
> > > > > > > > > | :--- | :--- |
> > > > > > > > > | FC | 26.56 |
> > > > > > > > > | CHTs (BRF) | 28.12 |
> > > > > > > > > | CHTs (DNM) | 27.62 |
> > > > > > > > >
> > > > > > > > > These results show that, when plugged into a strong DST framework, DNM consistently outperforms the BRF initialization and yields a perplexity closer to the fully connected baseline, despite using only 30% of the weights. This suggests that DNM can serve as a competitive drop-in replacement for existing sparse initializers in modern LLM training pipelines.
> > > > > > > > >
> > > > > > > > > Finally, we note that several hardware vendors are actively developing support for accelerating unstructured sparsity in training and inference (e.g., specialized sparse tensor cores and accelerators) (https://www.cerebras.ai/blog/harnessing-the-power-of-sparsity-for-large-gpt-ai-models). As such hardware becomes widely available, we expect unstructured sparse training to move closer to the mainstream. In that regime, a robust and general sparse initialization such as DNM can be readily adopted as a component of state-of-the-art architectures and DST methods, making its impact more practical for the broader community.

---

> > > > > > > > > > ### Author Response · Authors · 2025-12-03
> > > > > > > > > >
> > > > > > > > > > **Question 1:**
> > > > > > > > > >
> > > > > > > > > > What is the architecture of the models for MNIST? How many parameters are in the models? How many parameters are in the sparse models vs the fully connected models?
> > > > > > > > > >
> > > > > > > > > > We thank the reviewer for this observation. To address this topic, we will add the followingsubsection in the Appendix Hyperparameter Settings and Implementation Details.
> > > > > > > > > >
> > > > > > > > > > _Models are trained for 100 epochs using Stochastic Gradient Descent (SGD) with a learning rate of 0.025, a batch size of 32, and a weight decay of 5x10-4._
> > > > > > > > > >
> > > > > > > > > > **_Hyperparameters of MLP on Image Classification Tasks_**
> > > > > > > > > >
> > > > > > > > > > | Hyper-parameter | MLP |
> > > > > > > > > > | :--- | :--- |
> > > > > > > > > > | Hidden Dimension | 1568 (3072 for CIFAR10) |
> > > > > > > > > > | # Hidden layers | 3 |
> > > > > > > > > > | Batch Size | 32 |
> > > > > > > > > > | Training Epochs | 100 |
> > > > > > > > > > | LR Decay Method | Linear |
> > > > > > > > > > | Start Learning Rate | 0.025 |
> > > > > > > > > > | End Learning Rate | ${2.5e^{-4}}$ |
> > > > > > > > > > | zeta (fraction of removal) | 0.3 |
> > > > > > > > > > | Update Interval (for DST) | 1 epoch |
> > > > > > > > > > | Momentum | 0.9 |
> > > > > > > > > > | Weight decay | 5e-4 |
> > > > > > > > > >
> > > > > > > > > > _All sparse models are trained at a 99% sparsity level. For dynamic methods, we used SET, RigL, and CHTs. The regrowth strategy for CHTs is CH2_L3n (Muscoloni et al., 2018). For our DNM, we conduct a grid search over its key hyperparameters. We tested a mean dendrite count (M) of 3. For the dendritic, degree, receptive field width, and synaptic distributions, we searched across fixed, spatial Gaussian, and spatial inverted Gaussian options. The mean receptive field width (alpha) was fixed at 1.0. For the BSW baseline, the rewiring probability is searched in the set {0.0, 0.2, 0.4, 0.6, 0.8, 1.0}. For the BRF baseline, we searched the randomness parameter r over the same set of values, and also tested both fixed and uniform degree distributions._

---

> > > > > > > > > > > ### Author Response · Authors · 2025-12-03
> > > > > > > > > > >
> > > > > > > > > > > **Question 2: what is the evaluation strategy for the DNM models and other baselines on MNIST? how many different DNM variants are tested? How is the best model selected — on a held-out validation set? It could be useful to see a table of training, validation, and test accuracies for the different models tested.**
> > > > > > > > > > >
> > > > > > > > > > > We thank the reviewer for this crucial observation regarding the evaluation protocol. We acknowledge that our initial MLP experiments utilized the exact codebase from a previous work, which defaulted to reporting test accuracy directly. To rectify this and ensure rigorous evaluation, we have re-implemented the experimental pipeline for all MLP models. Our revised strategy is as follows:
> > > > > > > > > > >
> > > > > > > > > > > - Data Splitting: We partition the standard training set to create a strict, held-out validation set of 1,000 samples. The remaining samples are used for training.
> > > > > > > > > > >
> > > > > > > > > > > -Model Selection: We do not use the test set for hyperparameter tuning or checkpoint selection. During training, we evaluate the model on the validation set at the end of every epoch. We select and save the model checkpoint that achieves the highest accuracy on the validation set.
> > > > > > > > > > >
> > > > > > > > > > > - The reported test accuracy in our revised tables corresponds to the performance of this validation-selected model.
> > > > > > > > > > >
> > > > > > > > > > > The results are shown in response to your Weakness n.1. We note that due to our time limitations, we decided to omit CHTss from our updated manuscript. We decided to sacrifice this model for two reasons:
> > > > > > > > > > >
> > > > > > > > > > > 1. It didn’t add any extra insights, since its regrowth strategy is the same as CHTs.
> > > > > > > > > > >
> > > > > > > > > > > 2. It was initialized at a higher density, meaning that its reliance on the network initialization was weaker.
> > > > > > > > > > >
> > > > > > > > > > > Regarding the DNM variants, we performed a comprehensive topological grid search to evaluate the robustness of our method. We tested 81 distinct topological configurations per dataset (averaged over 3 random seeds). This grid resulted from permuting 3 distributions (Fixed, Spatial Gaussian, Spatial Inverse Gaussian) across 4 structural parameters: Dendritic distribution, Degree distribution, Receptive Window Width Distribution, Synaptic distribution. Finally, to demonstrate the stability of this approach, we provide the training dynamics of our best-performing MNIST model below.
> > > > > > > > > > >
> > > > > > > > > > > | Rank | Mean Val | Seed 1 (Val / Test) | Seed 2 (Val / Test) | Seed 3 (Val / Test) |
> > > > > > > > > > > | :--- | :--- | :--- | :--- | :--- |
> > > > > > > > > > > | 1 | 98.6000 | 98.3000 / 97.9900 | 99.0000 / 98.2500 | 98.5000 / 97.9800 |
> > > > > > > > > > > | 2 | 98.5333 | 98.3000 / 98.1900 | 98.6000 / 98.0500 | 98.7000 / 98.1000 |
> > > > > > > > > > > | 3 | 98.4667 | 98.4000 / 98.1500 | 98.6000 / 98.0100 | 98.4000 / 97.8900 |
> > > > > > > > > > > | 4 | 98.4667 | 98.0000 / 98.1000 | 98.6000 / 97.9300 | 98.8000 / 98.0000 |
> > > > > > > > > > > | 5 | 98.4000 | 98.4000 / 98.0300 | 98.2000 / 98.0900 | 98.6000 / 98.1400 |
> > > > > > > > > > >
> > > > > > > > > > > But to clarify, for the Transformer experiments, we select the checkpoint that achieves the highest next-token prediction accuracy on a held-out validation set, and then report BLEU on the test set using this checkpoint.
> > > > > > > > > > >
> > > > > > > > > > > **Question 3: Why are the error levels so low for the MNIST and other experiments? Is there other work that finds similar error levels when training different initializations on MNIST?**
> > > > > > > > > > >
> > > > > > > > > > > As mentioned, Zhang et al. (2025) (https://arxiv.org/abs/2501.19107) reported results with comparable standard error ranges. We performed tests on static sparse training for image classification at 99% sparsity on MNIST with the DNM initialization to prove that there is a correlation between the learning rate of our training and the standard errors between different seeds. The image found at: https://anonymous.4open.science/r/ICLR2026-3B7F/lrs.png shows that when adopting a learning rate of 0.025 we obtain stable results, where the standard error rarely surpasses 0.1.

---

### Official Review · Reviewer_Dxqp · 2025-11-01

**Soundness:** 2
**Presentation:** 1
**Contribution:** 2
**Rating:** 2
**Confidence:** 3

**Summary:**

This paper introduces the Dendritic Network Model (DNM), a generative framework for initializing sparse neural networks using biologically inspired connectivity principles. The model is evaluated across image classification and translation benchmarks and compared with several static and dynamic sparse training (DST) methods. DNM is shown to consistently outperform baselines at high sparsity levels and performs competitively with data-informed methods such as SNIP and CSTI.

**Strengths:**

The topic is timely and relevant: understanding the link between biological structure and sparse AI architectures is of broad interest to the ICLR community.

The model is novel as a generative topological initialization approach and appears to integrate concepts from network science and computational neuroscience.

The paper includes comprehensive experiments across multiple architectures and datasets, including transformer-based translation experiments in addition to image benchmarks.

**Weaknesses:**

The biological connection is not clearly articulated. The work lacks a discussion of DNM’s relationship to what is known about biological dendritic connectivity patterns. There has been significant progress here as a result of connectomics datasets (e.g. the MICrONs dataset and related papers) and it is a weakness that DNMs are not contextualised better here.

The paper’s clarity is a major limitation. Key terms and references to other methods are often introduced without adequate explanation or focus. For example, CHTs, SET and RigL are not adequately introduced and motivated. Why are they good comparisons to DNMs? Section 3 mixes implementation detail and inspiration in a way that is difficult to follow. Figure 2 is not motivated and described well enough. What is the purpose of the characterisation? It is not clear what the reader should take away from this figure, other than that the DNM can generate a range of networks.

Experimental methodology isn’t clearly communicated to the reader and is underexplained. For example, learning rate hyperparameter search doesn’t seem to have been done for MLPs, instead all models were trained with the same learning rate (Appendix B1).  This seems unlikely given the range of image datasets trained on.

Section 5 does analyse somewhat the structure of best performing DNM models. But this analysis is descriptive and speculative, and doesn’t clearly communicate the principles the best performing DNM models are leveraging for good task performance.

Minor:
- The term “sandwich layer” (L137) is unconventional and confusing; “hidden layer” would suffice unless a specific distinction is intended.

- Figure 4 is referenced in the main text but only appears in the appendix.

- Several claims (e.g., about scale-free structure, L252) require citations or clearer definitions.

**Questions:**

Line 51: Could you clarify the claimed relationship between dendritic computation and convolutional models?

Line 40-43: The efficiency statements should either be framed as hypotheses or supported by explicit citations.

Line 57: How do you operationally define “dendritic topology”?

Figure 4 is an appendix figure.

Line 211: Please describe how the coalescent embedding in hyperbolic space is computed and what it reveals about network structure.

Line 252: Claim about scale free networks should be cited.

---

> ### Author Response · Authors · 2025-12-03
>
> **Weakness n.1: The biological connection is not clearly articulated. The work lacks a discussion of DNM’s relationship to what is known about biological dendritic connectivity patterns. There has been significant progress here as a result of connectomics datasets (e.g. the MICrONs dataset and related papers) and it is a weakness that DNMs are not contextualised better here [REPLY 1 of 3]**:
> Thank you for this insightful feedback. To address the review’s concern, we expanded the Introduction section to discuss how large-scale datasets like MICrONS are providing an unprecedented, data-driven understanding of dendritic connectivity patterns, such as synaptic clustering and non-uniform random receptive field organizations. In particular, we clarify that the DNM is proposed as a generative, topological abstraction of dendritic principles, which is indeed able to tailor different variants of non-uniform random receptive field organizations. Rather than a direct emulation of  specific biological circuits, this is a dendritic-inspired design of bipartite network connectivity between a soma layer and a synaptic input layer. We now explicitly discuss how this connectomics data provides a descriptive blueprint of these principles. We then position our work as a necessary generative framework that translates these high-level principles, such as modularity, hierarchy, and clustered connectivity, into a controllable spectrum of initializable topologies for ANNs, a gap not currently filled by descriptive connectomics or more focused computational neuron models.
> We have reviewed the Introduction as follows:

---

> ### Author Response · Authors · 2025-12-03
>
> **Weakness n.1: The biological connection is not clearly articulated. The work lacks a discussion of DNM’s relationship to what is known about biological dendritic connectivity patterns. There has been significant progress here as a result of connectomics datasets (e.g. the MICrONs dataset and related papers) and it is a weakness that DNMs are not contextualised better here [REPLY 2 of 3]:**
>
> _Artificial neural networks (ANNs) have demonstrated remarkable potential in various fields; however, their size, often comprising billions of parameters, poses challenges for both economic viability and environmental sustainability. Biological neural networks, in contrast, can efficiently process information using ultra-sparse structures (Drachman, 2005; Walsh, 2013). This efficiency arises from the brain's highly structured and evolutionarily optimized network topology. A central component of this architecture is the dendritic organization, the primary receptive surface of the neuron (Cuntz et al., 2010). Conventional ANNs omit a crucial component of the brain's efficiency since they traditionally depict neurons as simple point-like integrators, mainly ignoring the computing power inherent in the intricate structure of dendrites._
>
> _Research has revealed that dendrites are not passive conductors but active computational units capable of performing sophisticated, nonlinear operations and integrations  (London & Hausser, 2005; Larkum, 2013). This insight has motivated theoretical frameworks that model a single neuron as a multi-layer network, where dendritic branches act as nonlinear subunits that feed a final integrator at the cell body (Lauditi et al., 2025). As clarified by recent works on dendritic artificial neural networks (Li et al., 2020; Chavlis & Poirazi, 2025), the dendritic tree's ability to sample and nonlinearly integrate restricted parts of the input space can be used in neuromorphic physical networks  (Li et al., 2020) or in artificial convolutional layers (Chavlis & Poirazi, 2025). In this paradigm, distinct dendritic branches process specific, localized receptive fields without sharing weights, allowing for precise, location-specific feature integration. Additional efforts to translate these principles into neuromorphic systems have confirmed that also dendritic morphology has a significant impact on spatio-temporal processing and performance (Baek et al., 2024; Jones & Kording, 2021). These approaches, however, are often limited to fixed structures that mimic the computational non-linearity or direct morphology of biological neurons, overlooking the broader rules of connectivity, which also includes non uniform random network dinamic rewiring by synaptic turnover (Frank et al., 2018; Zhang et al., 2025)._
>
> _Our understanding of these topological constraints has been revolutionized by the advent of large-scale functional connectomics (mic, 2025). These studies reveal that biological neural networks are not uniformly randomly wired; rather, they exhibit specific, non-uniform random connectivity patterns characterized by  "like-to-like" wiring rules and distinct structural motifs across cortical layers. Translating these high-level connectomic principles, such as modularity, hierarchy, and non-random receptive field organization, into scalable, topological generative frameworks for designing the sparse structure of artificial networks remains an open challenge._
>
> _To address the gap for a flexible, principled framework for generating and testing dendritic topologies, we introduce a dendritic-inspired network science generative model for sparse topology design of bipartite layers in deep neural networks: the Dendritic Network Model (DNM). The novelty of DNM lies in the elaborated mechanism to model the non-uniform topological organization of the receptive fields on the input layers. The model's parametric approach enables the systematic exploration of the relationship between network structure and computational function. The DNM provides a principled method for generating sparse network initializations that can be integrated into modern deep learning frameworks. We demonstrate that this approach can improve performance over standard sparse initialization techniques and offers a powerful platform for exploring how structural constraints, inspired by biology, can lead to more efficient and capable artificial neural networks._

---

> > ### Author Response · Authors · 2025-12-03
> >
> > **Weakness n.1: The biological connection is not clearly articulated. The work lacks a discussion of DNM’s relationship to what is known about biological dendritic connectivity patterns. There has been significant progress here as a result of connectomics datasets (e.g. the MICrONs dataset and related papers) and it is a weakness that DNMs are not contextualised better here [REPLY 3 of 3]:**
> >
> > _Our approach can be contrasted with other dendritic-inspired methodologies in the field. For instance, Li et al. (2020) experimentally demonstrated a fully integrated hardware network using memristor devices, where artificial dendrites provided non-linear integration and filtering to achieve highly efficient physical networks. Subsequently, Malakasis et al. (2023) utilized bio-realistic spiking neural networks to show how active dendrites combined with uniformly random synaptic turnover can optimize learning in binary classification scenarios. These works pave the way for recent advancements like the work of Chavlis et al (2025), which presents a dendritic-emulating model that reproduces the nonlinear integrative functions of dendrites. In their framework, a dendrite is mapped to a node within a tree-like subnetwork, creating a nonlinearcomputational component for larger networks. Contrarily to these methods, our DNM is merely a brain-inspired network science model and a compact methodology to initialize the topology between two layers in a way that is dendritic-inspired. The dendrites in our model emerge from the topological clustering of synaptic connections, and our goal is to demonstrate that organizing connectivity according to these principles can yield performance gains. Thus, while Li et al. (2020) and Chavlis et. al (2025) focus on the emulation of nonlinear processing in tree-like dendritic subnetwork structures, our work investigates topological dendritic-inspired principles that allows modelling the initial sparse connectivity organization from a network science perspective._
> >
> > _In this article, we describe the Dendritic Network Model in detail and analyse its topology and geometric characterization. We evaluate its effectiveness with extensive experiments across multiple architectures and tasks. To assess its basic functionality, we use it to initialize several static and dynamic sparse training (DST) methods on MLPs for image classification on the MNIST (LeCun et al., 2002), EMNIST (Cohen et al., 2017), Fashion MNIST (Xiao et al., 2017), and CIFAR-10 (Krizhevsky,2009) datasets. The results show that DNM clearly outperforms other sparse initialization methods over all training models tested at 99% sparsity. Next, we extend the tests on Transformers (Vaswani et al., 2017a) for Machine Translation on the Multi30k en-de (Elliott et al., 2016), IWSLT14 en-de (Cettolo et al., 2014), and WMT17 en-de (Bojar et al., 2017) benchmarks. On this architecture, DNM outperforms all topological initialization methods at high sparsity levels. These findings underscore the potential of DNM in enabling highly efficient and effective network initialization for large-scale sparse neural network training. By analyzing the best-performing DNM topologies, we can also gain insights into the relationship between network geometry, data structure, and model performance._

---

> ### Author Response · Authors · 2025-12-03
>
> **Weakness 2: The paper’s clarity is a major limitation. Key terms and references to other methods are often introduced without adequate explanation or focus. For example, CHTs, SET and RigL are not adequately introduced and motivated. Why are they good comparisons to DNMs? Section 3 mixes implementation detail and inspiration in a way that is difficult to follow. Figure 2 is not motivated and described well enough. What is the purpose of the characterisation? It is not clear what the reader should take away from this figure, other than that the DNM can generate a range of networks. [REPLY 1/2]:**
>
> We thank the reviewer for highlighting these clarity limitations. The reviewer asks why these methods are "good comparisons to DNMs." DNM is not a competing dynamic sparse training (DST) framework, but an initialization method that provides the starting topology for these frameworks. CHTs, SET, and RigL are the three representative DST methods that use three fundamentally different ways to grow new links: uniformly random (SET), via gradient-based information (RigL), and through the network science-based Cannistraci-Hebb Theory for gradient-free link regrowth. By evaluating our DNM-initialized topology on these three diverse DST frameworks, we can demonstrate that its structural advantage is robust and not an artifact of one specific training or regrowth method. This allows us to isolate the benefit of the initial topology. To clarify this aspect, we have restructured section 4.1 to separate the "Testbeds" (DST frameworks) from the "Comparisons" (Initialization baselines):
>
> _**Experimental Setup**_
> _To evaluate the structural advantage of DNM over other initialization methods, we conduct experiments over two regimes: Static Sparse Training and Dynamic Sparse Training (DST)._
>
> **_Static Sparse Training_**
> _We first evaluate DNM on Multilayer Perceptrons (MLPs) for image classification tasks on the MNIST (LeCun et al., 2002), Fashion MNIST (Xiao et al., 2017), EMNIST (Cohen et al., 2017), and CIFAR10 (Krizhevsky, 2009) datasets. In this regime, the topology remains fixed after initialization to isolate the performance of initial sparse network._
>
> **_Dynamic Sparse Training (DST)_**
> _To validate the robustness of DNM as an initialization strategy for evolving topologies, we integrate it into state-of-the art DST frameworks. We select three DST methods that represent different landscapes of topological evolution mechanisms: SET (Mocanu et al., 2018) utilises random link regrowth; RigL (Evci et al., 2020) adopts gradient-based link regrowth; CHTs (Zhang et al., 2025) uses a network science-based Hebbian-inspired  gradient-free link regrowth. A detailed description of these models is provided in Appendix H.2. By testing DNM on these fundamentally different regrowth strategies, we aim to prove that the benefits of the initialization are not limited to a specfic training paradigm. Finally, we apply DNM to Transformer (Vaswani et al., 2017b) models for machine translation tasks on the Multi30k en-de (Elliott et al., 2016), IWSLT14 en-de (Cettolo et al., 2012), and WMT17 (Bojar et al., 2017) datasets._
>
> **_Implementation details_**
> _For MLP training, we sparsify all layers except the final layer to prevent disconnected output neurons, noting that the final layer has relatively fewer connections compared to previous layers. Comprehensive parameter settings are detailed in Appendix D, and sensitivity tests on the hyperparameters of the DNM are provided in Appendix E._
>
> **_Baseline methods_**
> _We compare the performance of the DNM initialization against baseline topologies found in the literature. On static sparse training, we compare DNM with a randomly initialized network, the Bipartite Small World (BSW) (Zhang et al., 2024B), the Bipartite Receptive Field (BRF) (Zhang et al., 2025), the Ramanujan graph (Kalra et al.) initialization techniques, the RadiX-Nets (Kepner & Robinett, 2019) and dANN-R (Chavlis & Poirazi, 2025). We also include CSTI (Zhang et al., 2024a) and SNIP (Lee et al., 2018) as the baseline models, noting that their comparison is inherently unfair due to their data-informed nature. For dynamic sparse training, we compare DNM with a random initialization, BRF, BSW, Ramanujan graph, RadiX-Nets, three variants of dANNs, SNIP and CSTI. Finally, for tests on Transformer models, we compare DNM with the BRF initialization, which was proven to be the state-of-the-art sparse initialization method in previous studies (Zhang et al., 2025)._

---

> > ### Author Response · Authors · 2025-12-03
> >
> > **Weakness 2: The paper’s clarity is a major limitation. Key terms and references to other methods are often introduced without adequate explanation or focus. For example, CHTs, SET and RigL are not adequately introduced and motivated. Why are they good comparisons to DNMs? Section 3 mixes implementation detail and inspiration in a way that is difficult to follow. Figure 2 is not motivated and described well enough. What is the purpose of the characterisation? It is not clear what the reader should take away from this figure, other than that the DNM can generate a range of networks. [REPLY 2/2]:**
> >
> > We understand the reviewer’s concern regarding Section 3’s clarity. To fix this, we have restructured Section 3 to create a much clearer and more logical progression for the reader:
> > - Section 3.1 (Biological Inspiration) remains as the conceptual foundation .
> > - We have added a new Section 3.2, "The DNM Generative Algorithm," which provides the high-level, step-by-step description of how a network is built . This content was previously located in the Appendix.
> > - The original Section 3.2 ("Parametric Specification") is now Section 3.3. This section now logically follows the algorithm it parameterizes.
> > - The original Section 3.3 ("Network Topology and Geometric Characterization") is now Section 3.4 , which analyzes the outputs of the previously described algorithm and parameters.
> > This new structure creates a much cleaner flow: from the biological idea (3.1), to the algorithm that models it (3.2), to the parameters that control the algorithm (3.3), and finally to the resulting topologies (3.4). Here is the newly created Section 3.2:
> >
> > **_The DNM Generative Algorithm_**
> > _To translate these biological principles into a computational structure, the DNM produces the sparse connectivity matrix of sandwich layers via a generative algorithm. The process builds connections iteratively for each output neuron j through the following steps: (1) Degree determination: first, determine the total degree for the output neuron based on a specific degree distribution strategy; (2) Receptive field definition: define the receptive field for the output neuron by topologically mapping the output neuron’s position to a central point on the input layer and establishing a receptive window around this center; (3) Dendritic allocation: determine the number of dendritic branches used to connect the output neuron to the input layer; (4) Dendritic Placement: place evenly spread-out dendritic brances within the neuron’s receptive window defined in step 2; (5) Synaptic distribution: distribute the output neuron’s total degree across the dendrites. Appendix F describes the algorithm in depth._
> >
> > We understand the reviewer’s point, and to address it we have augmented the introductory paragraph of this section (now Section 3.4) to explicitly state the goal of this characterization. Our paper's fundamental hypothesis is that a network’s initial topology provides a structural advantage in learning. To test this, we must first demonstrate that our DNM is a flexible generative framework capable of producing a diverse and controllable landscape of network topologies. Therefore, the purpose of Figure 2 is to provide quantitative and visual proof of this flexibility. We show that by adjusting DNM’s parameters, we can controllably generate architectures with vastly different properties, from unstructured random graphs (Fig. 2a) to highly modular (Fig. 2b) and even scale-free, hierarchical ones (Fig. 2c) . This controllable range is precisely what enables our later analysis (in Section 5) to explore the critical link between network structure and task performance. Here is the revised introductory paragraph of Section 3.4:
> >
> > **_Network Topology and Geometric Characterization_**
> >
> > _A central hypothesis of this work is that specific topological features, such as modularity and hierarchy, confer distinct inductive biases that facilitate learning in ANNs. To test this hypothesis, it is essential to demonstrate that the DNM is not limited to a single structural configuration but rather functions as a flexible generative framework capable of accessing a diverse landscape of topologies. In this section, we systematically vary the hyperparameters defined in Section 3.3 to characterize this landscape. Our goal is to show that by tuning the DNM’s parameters, we can controllably transition the network architecture across three distinct regimes: from unstructured uniformly connected random graphs, to input-order-dependent highly modular networks, and finally to input-order-dependent hierarchical, scale-free (Barabasi & Albert, 1999) topologies._

---

> > > ### Author Response · Authors · 2025-12-03
> > >
> > > **Weakness 3: Experimental methodology isn’t clearly communicated to the reader and is underexplained. For example, learning rate hyperparameter search doesn’t seem to have been done for MLPs; instead, all models were trained with the same learning rate (Appendix B1). This seems unlikely given the range of image datasets trained on.**
> > >
> > > We thank the reviewer for raising this valid concern about our experimental methodology. For the MLP image classification experiments, we used learning rate decay strategy of a starting learning rate 0.025 and then decay to 0.00025. We acknowledge the reviewer's insightful point that a single learning rate is unlikely to be optimal for all four datasets and all initialization methods. To address this and further validate the robustness of our findings, we conducted a new learning rate sensitivity analysis on a restricted hyperparameter space for the static sparse training MLP experiments on MNSIT, EMNIST, and Fashion MNIST. We report the results. The results can be found at the images below:
> > > - MNIST: https://anonymous.4open.science/r/ICLR2026-3B7F/lrs.png
> > > - Fashion MNIST: https://anonymous.4open.science/r/ICLR2026-3B7F/lrs_fmnist.png
> > > - EMNIST: https://anonymous.4open.science/r/ICLR2026-3B7F/lrs_emnist.png  We commented the figures in a newly created appendix section (Appendix O):
> > >
> > > _This section analyzes the impact of learning rate variations on model stability (Standard Error) and predictive performance (Mean Test Accuracy) across three distinct datasets: Fashion MNIST, EMNIST, and MNIST. The plots illustrate the trade-off between convergence speed and stability as the learning rate is increased from 0.01 to 0.1 (Figure 9). Across all three datasets, a learning rate in the range of 0.025 to 0.05 appears to be the optimal operating window. This range consistently provides the best balance of maximizing test accuracy while avoiding the instability and divergence (high standard error) associated with rates approaching 0.10._

---

> > > > ### Author Response · Authors · 2025-12-03
> > > >
> > > > **Weakness 4: Section 5 does analyse somewhat the structure of best performing DNM models. But this analysis is descriptive and speculative, and doesn’t clearly communicate the principles the best-performing DNM models are leveraging for good task performance.**
> > > >
> > > > We thank the reviewer for this constructive feedback. We first wish to highlight that the Dendritic Network Model is explicitly designed to traverse a diverse topological landscape. As described in Section 3.4, by systematically varying the generative hyperparameters, specifically the dendritic count (M), receptive field width (alpha), and their spatial distributions, we can precisely control the resulting topological metrics. This allows us to generate networks ranging from clustered, modular architectures to scale-free hierarchies. To address the lack of a sufficient quantitative, evidence-based description of the topological characteristics of the best-performing DNM models, we developed a deeper analysis to move from speculation to a clear explanation of the principles our models are leveraging. We dedicated a new appendix section (Appendix N) named Relationship between Topology and Performance where we deeply analyse the correlation between performance and the topological measures of four different image classification datasets tested. We paste the section below:
> > > >
> > > > _In this section, we analyze how different topological properties of DNM-initialized networks correlate with their performance on image classification tasks. We focus on four key graph-theoretic metrics: powerlaw distribution, average path length, clustering coefficient, and degree distribution. Understanding these relationships can provide insights into why certain configurations yield better results. To investigate the structural drivers of performance in DNM-initialized networks, we analyzed the correlation between key graph-theoretic metrics and test accuracy across the MNIST, Fashion-MNIST, EMNIST, and CIFAR-10 datasets. For each dataset, we isolated the top 10, middle 10, and bottom 10 performing models from our hyperparameter search to visualize how topological variance dictates model efficacy. The results are summarized in Figure 8._
> > > >
> > > > _**Powerlaw gamma**_ _Regarding the degree distribution exponent (γ), we see Fashion-MNIST aligning with CIFAR-10. Both datasets exhibit a positive correlation between accuracy and γ (r = 0.31 for Fashion-MNIST, r = 0.45 for CIFAR-10). Since a higher γ implies a steeper decay in the degree distribution (fewer extreme hubs), this indicates that complex tasks prefer a more distributed connectivity where information processing is shared among many nodes rather than concentrated in a few central hubs. Conversely, MNIST and EMNIST show negative correlations (r = −0.16 and r = −0.27), suggesting they benefit from the strong, centralized integration provided by heavy-tailed, hub-dominated topologies._
> > > >
> > > > _**Structural Consistency**_ _Across all four datasets, we observe a consistent negative correlation between Structural Consistency (σc) and accuracy. Fashion-MNIST exhibits the strongest negative correlation (r = −0.81), reinforcing the finding that strict structural predictability is detrimental to initialization performance, regardless of the dataset complexity._
> > > >
> > > > _**Modularity**_ _We observe that higher modularity (Q) correlates positively with performance across CIFAR-10. The best-performing topologies exhibit high modularity, suggesting that complex natural image recognition benefits from distinct, specialized communities of neurons that process local features independently before integration. Conversely, for the simpler MNIST and EMNIST datasets, lower modularity appears advantageous, indicating that a more integrated network structure suffices for less complex tasks._
> > > >
> > > > _**Characteristic Path**_ _Like CIFAR-10, Fashion-MNIST models show a positive correlation (r = 0.39), indicating a benefit from longer path lengths that preserve local processing. This stands in contrast to MNIST and EMNIST (r < 0), which favor ”small-world” architectures with short global integration paths._
> > > >
> > > > Figures 8 (a), (b), (c), and (d) can be found at the following links:
> > > > - (a) https://anonymous.4open.science/r/ICLR2026-3B7F/mnist.png
> > > > - (b) https://anonymous.4open.science/r/ICLR2026-3B7F/emnist.png
> > > > - (c) https://anonymous.4open.science/r/ICLR2026-3B7F/fmnist.png
> > > > - (d) https://anonymous.4open.science/r/ICLR2026-3B7F/cifar.png

---

> > > > > ### Author Response · Authors · 2025-12-03
> > > > >
> > > > > **Weakness 5: The term “sandwich layer” (L137) is unconventional and confusing; “hidden layer” would suffice unless a specific distinction is intended.**
> > > > >
> > > > > We thank the reviewer for pointing out this potential source of confusion. We agree that the term "sandwich layer" is unconventional, and we appreciate the opportunity to clarify the specific distinction we intended.
> > > > > The reviewer is correct that "hidden layer" typically refers to the neurons (nodes) that are not in the input or output layers. In our work, however, we must distinguish the neurons from the connections between them, as DNM is a generative model for network topology (i.e., the edges).
> > > > > We use the term "sandwich layer" to refer specifically to the bipartite subnetwork of the neurons (two hidden layers) and the connections (edges) that exists between them, as written in line 137 of the original manuscript: _(a bipartite subnetwork that links the input nodes of a layer to their output nodes)_. This distinction is critical because our model is generating, not only the layers of neurons themselves, but also the set of links between them.
> > > > > We recognize this term was not clearly defined. We revise Section 3.1 to explicitly clarify this terminology and our rationale for using it, as follows:
> > > > >
> > > > > _The architecture of the Dendritic Network Model (DNM) is inspired by the structure of biological neurons. In the nervous system, neurons process information through complex, branching extensions called dendrites, which act as the primary receivers of synaptic signals. Inspired by this phenomenon, the DNM imposes a structured, dendrite-like organization on how output neurons connect to the preceding layer's inputs (Figure 1). In this work, we refer to the neurons of two adjacent layers and the set of connections between them as a sandwich layer, a term already used in prior literature to denote the bipartite subnetwork of edges that lies between one layer of neurons and the next (https://www.preprints.org/manuscript/202207.0139). We retain this terminology because it captures the specific object our model generates, the pattern of topology, rather than the neurons themselves. By contrast, the term hidden layer refers to the neurons in intermediate layers. Since the DNM defines how neurons connect, but does not generate or modify the neurons themselves, ``hidden layer" would not accurately describe the structural entity under consideration. Within each sandwich layer, each output neuron forms multiple dendritic branches, where each branch connects the neuron to a contiguous block of input neurons. These blocks are separated by inactive spaces, segments of the input layer to which the neuron does not connect. All branches belonging to a given output neuron must lie within a predefined local receptive window, resulting in a structured, compartmentalized connectivity pattern. This design moves away from unstructured random sparsity and instead emulates the localized, clustered organization characteristic of biological dendrites._

---

> > > > > > ### Author Response · Authors · 2025-12-03
> > > > > >
> > > > > > **Weakness 6: Figure 4 is referenced in the main text but only appears in the appendix.**
> > > > > >
> > > > > > We thank the reviewer for spotting this oversight. We have corrected the main text to reference the newly created Appendix B rather than citing Figure 4 directly. This directs the reader to the appropriate section where the figure and its corresponding analysis are located.

---

> ### Author Response · Authors · 2025-12-03
>
> **Weakness 7: Several claims (e.g., about scale-free structure, L252) require citations or clearer definitions.**
>
> We thank the reviewer for this helpful comment. To ensure all claims are properly supported, we will add the appropriate citation (Barabasi, 1999) for the scale-free structure claim to the main text at L252.
> We appreciate the reviewer's emphasis on clear definitions. As noted, we have provided comprehensive definitions for all key network science terms used, including "scale-free," in the Appendix. Moreover, I have made the following edits to further address the reviewer’s concerns:
>
> - **Before** A key finding is that by setting a spatial Gaussian degree distribution (Figure 3c), the DNM generates a hierarchical structure where gamma=2.30, typical for scale-free (Barabasi, 1999) networks. This is evidenced by the tree-like structure in the hyperbolic embedding.
>
> - **Now** A key finding is that by setting a spatial Gaussian degree distribution (Figure 2c), the DNM generates a hierarchical topology that exhibits scale-free (Barab´asi & Albert, 1999) properties. Specifically, the resulting degree distribution follows a power law $$P(k)∼k^{−γ}$$ with an exponent γ = 2.30. Since typical scale-free networks exhibit 2<γ< 3 (Barabasi & Albert, 1999), this confirms that DNM can synthesize architectures with hub-like characteristics and hierarchical organization purely through parametric initialization.
>
> - We added the following to the preamble of section 3.3 for better clarity:
> To apply biological spatial principles to non-spatial MLPs, we index neurons i ∈ {1, …, N}, and their physical location xi is defined linearly such that the distance between adjacent indices is minimized. This allows us to define "spatial" distributions where connectivity probabilities depend on the relative distance |xi - xj| between neurons in adjacent layers.
>
> - I added this to section 3.3 under the definition of Synaptic distribution:
> While the degree distribution determines the total connectivity ${k_j}$ of an output neuron j, the synaptic distribution governs the partition of ${k_j}$ across the neuron's Mj dendritic branches. Formally, if ${s_{j,b}}$ is the number of synapses on the b-th dendritic branch of neuron j, the distribution ensures $$\sum_{b=1}^{M_j}s_{j,b} =k_j.$$ This allocation can be uniform (fixed), stochastic (non-spatial), or topology-dependent (spatial), allowing specific branches, such as those in the center of the receptive field, to be more densely connected than distal branches.
>
> - I added this to section 3.3 at the beginning of the definition of the Receptive Field Width Distribution:
>
> _The receptive field of an output neuron j is defined as the contiguous subset of input neurons to which j potentially connects. We define a mapping function ɸ(j) that projects the index of output neuron j to a center coordinate on the input layer. The receptive field is then the interval [ɸ(j) - Wj/2, ɸ(j) +Wj/2,]$, where Wj is the receptive field width determined by the parameter ɑ._

---

> > ### Author Response · Authors · 2025-12-03
> >
> > **Question 1: Could you clarify the claimed relationship between dendritic computation and convolutional models?**
> >
> > We thank the reviewer for highlighting this important relationship. We have revised the Introduction to explicitly address the parallels and distinctions between the Dendritic Network Model (DNM) and convolutional architectures. While both mechanisms utilize local receptive fields to process information, they differ fundamentally in how they handle parameters and topology.
> >
> > As clarified in our revision, the dendritic tree’s ability to sample restricted parts of the input space mirrors the locality principle of convolutional layers. However, unlike Convolutional Neural Networks (CNNs) which rely on weight sharing (applying the same filter across the input), DNM employs location-specific connectivity. In DNM, distinct dendritic branches process specific, localized receptive fields without sharing weights, allowing for precise, heterogeneous feature integration that forms complex network topologies (e.g., scale-free or small-world) rather than uniform grids.
> >
> > _As clarified by recent work on dendritic artificial neural networks (Chavlis & Poirazi, 2025), the dendritic tree's ability to sample restricted parts of the input space mirrors the operation of convolutional layers. In this paradigm, distinct dendritic branches process specific, localized receptive fields without sharing weights, allowing for precise, location-specific feature integration._

---

> > > ### Author Response · Authors · 2025-12-03
> > >
> > > **Question 2: The efficiency statements should either be framed as hypotheses or supported by explicit citations.**
> > >
> > > We thank the reviewer for this insightful comment regarding the framing of our efficiency claims. To address this, we have added a dedicated Limitations section to the manuscript.
> > >
> > > _**Limitations**_
> > >
> > > _While the Dendritic Network Model demonstrates significant improvements in accuracy at extreme sparsity levels, we acknowledge a practical limitation regarding current hardware acceleration. Most contemporary GPUs are highly optimized for dense matrix operations; consequently, unstructured sparsity does not always translate into training speedups without specialized software support. However, the hardware landscape is rapidly evolving to address this bottleneck. Emerging technologies are specifically designed to efficiently support unstructured sparsity \citep{lie2022harnessing}. Our method is designed to be future-proof, positioned to fully leverage dynamic sparse training and inference as this specialized hardware becomes widely accessible._

---

> > > > ### Author Response · Authors · 2025-12-03
> > > >
> > > > **Question 3: How do you operationally define “dendritic topology”?**
> > > >
> > > > We thank the reviewer for asking for this clarification. We had operationally defined dendritic topology in the original manuscript (Line 136), where we stated:
> > > > Inspired by this phenomenon, the DNM shapes each sandwich layer... in a way that output neurons are connected to distinct groups of input neurons with multiple dendritic branches... In the DNM, an output neuron’s connections are organized into distinct clusters called dendrites. Each branch connects the output neuron to a consecutive batch of input neurons... separated by 'inactive spaces'... All of an output neuron’s branches must be formed within a localized receptive window…
> > > > However, to make this operational definition more precise, we have rephrased Section 3.1 as follows:
> > > >
> > > > _The architecture of the Dendritic Network Model (DNM) is inspired by the structure of biological neurons. In the nervous system, neurons process information through complex, branching extensions called dendrites, which act as the primary receivers of synaptic signals. Inspired by this phenomenon, the DNM imposes a structured, dendrite-like organization on how output neurons connect to the preceding layer's inputs (Figure 1). In this work, we refer to the set of connections between two adjacent layers as a sandwich layer, a term already used in prior literature to denote the bipartite subnetwork of edges that lies between one layer of neurons and the next. We retain this terminology because it captures the specific object our model generates, the pattern of connections, rather than the neurons themselves. By contrast, the term hidden layer refers to the neurons in intermediate layers. Since the DNM defines how neurons connect, but does not generate or modify the neurons themselves, ``hidden layer" would not accurately describe the structural entity under consideration. Within each sandwich layer, each output neuron forms multiple dendritic branches, where each branch connects the neuron to a contiguous block of input neurons. These blocks are separated by inactive spaces, segments of the input layer to which the neuron does not connect. All branches belonging to a given output neuron must lie within a predefined local receptive window, resulting in a structured, compartmentalized connectivity pattern. This design moves away from unstructured random sparsity and instead emulates the localized, clustered organization characteristic of biological dendrites._

---

> > > > > ### Author Response · Authors · 2025-12-03
> > > > >
> > > > > **Question 4:Figure 4 is an appendix figure.**
> > > > >
> > > > > Already addressed in weekness 6
> > > > >
> > > > > **Question 5:.Please describe how the coalescent embedding in hyperbolic space is computed and what it reveals about network structure.**
> > > > >
> > > > > We appreciate the reviewer suggesting that we clarify the methodology and interpretation of the coalescent embedding. While the algorithmic details of the embedding remain in Appendix A, we have revised the subsection on Network Topology and Geometric Characterization to provide a concise operational definition of the embedding in the main text, as follows:
> > > > >
> > > > > _To visualize the network's latent geometry, the coalescent embedding maps nodes onto a 2D hyperbolic disk. In this representation, angular coordinates are computed via non-linear dimensionality reduction to cluster structurally similar nodes, while radial coordinates are derived from node popularity (degree). Consequently, this visualization reveals two key structural properties: hierarchy (nodes near the center act as hubs, while peripheral nodes are leaf-like) and modularity (angular grouping indicates community structure). The full algorithmic details are provided in Appendix A._

---

> > > > > > ### Author Response · Authors · 2025-12-03
> > > > > >
> > > > > > **Question 5: Please describe how the coalescent embedding in hyperbolic space is computed and what it reveals about network structure.**
> > > > > >
> > > > > > We appreciate the reviewer suggesting that we clarify the methodology and interpretation of the coalescent embedding. While the algorithmic details of the embedding remain in Appendix A, we have revised the subsection on Network Topology and Geometric Characterization to provide a concise operational definition of the embedding in the main text, as follows:
> > > > > >
> > > > > > _To visualize the network's latent geometry, the coalescent embedding maps nodes onto a 2D hyperbolic disk. In this representation, angular coordinates are computed via non-linear dimensionality reduction to cluster structurally similar nodes, while radial coordinates are derived from node popularity (degree). Consequently, this visualization reveals two key structural properties: hierarchy (nodes near the center act as hubs, while peripheral nodes are leaf-like) and modularity (angular grouping indicates community structure). The full algorithmic details are provided in Appendix A._

---

> > > > > > > ### Author Response · Authors · 2025-12-03
> > > > > > >
> > > > > > > **Question 6: Claim about scale free networks should be cited.**
> > > > > > >
> > > > > > > We thank the reviewer for spotting this oversight. We added the citation to Barabasi & Albert (1999) when we mentioned scale-free networks.

---

### Meta-Review · Area_Chair_PwDS · 2026-01-07

**Summary:**

Reviewers found the proposed Dendritic Network Model (DNM) to be interesting and timely, with a broad experimental evaluation showing consistent gains over existing sparse initialization methods at extreme sparsity. However, the paper initially suffered from substantial clarity and presentation issues, including unclear positioning of the biological inspiration, confusing structure, and underexplained experimental choices. Reviewers were also concerned that the analysis connecting topology to performance was largely descriptive and that some methodological details (e.g., learning-rate choices) were insufficiently justified, making it difficult to fully assess the strength of the claims.

**Reviewer Concerns:**

The rebuttal partially addresses most major concerns by clarifying the intended level of biological abstraction, restructuring the core sections for improved readability, adding methodological clarifications and sensitivity analyses, and extending the topology–performance analysis with additional quantitative evidence.

Some concerns remain only partially resolved, particularly regarding overall readability, the density of exposition, and whether the biological motivation is sufficiently concrete for all readers. In addition, some newly added experiments requested by reviewers (e.g., in reinforcement learning or other domains) are limited in scale and therefore insufficient to fully validate the generality of the approach. Finally, despite the clarifications, given the related-work concerns raised in the reviews, the contribution may still be perceived as incremental by some reviewers.

**Reviewer Scores:**

It is difficult to precisely estimate how individual reviewers would have updated their scores given the limited discussion. The two reviewers with the lowest initial scores (i.e., 0 and 2) would likely increase slightly, given that many of their criticisms were explicitly addressed in the rebuttal, although they would require probably a deeper discussion to be fully convinced. The two reviewers who were already neutral or mildly positive are less likely to change their scores (i.e., 6 and 6). Overall, a hypothetical reviewers-authors discussion would likely narrow the initial disagreement, but would not lead to a strong consensus shift. I encourage the authors to consider the received feedback in order to strengthen the next version of the paper.

---

### Decision · Program_Chairs · 2026-01-26

Reject